

# Addressing Internet of Things security by enhanced sine cosine metaheuristics tuned hybrid machine learning model and results interpretation based on SHAP approach

Milos Dobrojevic[1], Miodrag Zivkovic[1], Amit Chhabra[2], Nor Samsiah Sani[3], Nebojsa Bacanin[1] and Maifuza Mohd Amin[3]

[1] Informatics and Computing, Singidunum University, Belgrade, Serbia
[2] Department of Computer Engineering & Technology, Guru Nanak Dev University, Amritsar, India
[3] Center for Artificial Intelligence Technology, Universiti Kebangsaan Malaysia, Bangi, Selangor, Malaysia

Corresponding authors
Amit Chhabra, amit.cse@gndu.ac.in
Nor Samsiah Sani, norsamsiahsani@ukm.edu.my

## ABSTRACT

An ever increasing number of electronic devices integrated into the Internet of Things (IoT) generates vast amounts of data, which gets transported *via* network and stored for further analysis. However, besides the undisputed advantages of this technology, it also brings risks of unauthorized access and data compromise, situations where machine learning (ML) and artificial intelligence (AI) can help with detection of potential threats, intrusions and automation of the diagnostic process. The effectiveness of the applied algorithms largely depends on the previously performed optimization, *i.e.*, predetermined values of hyperparameters and training conducted to achieve the desired result. Therefore, to address very important issue of IoT security, this article proposes an AI framework based on the simple convolutional neural network (CNN) and extreme machine learning machine (ELM) tuned by modified sine cosine algorithm (SCA). Not withstanding that many methods for addressing security issues have been developed, there is always a possibility for further improvements and proposed research tried to fill in this gap. The introduced framework was evaluated on two ToN IoT intrusion detection datasets, that consist of the network traffic data generated in Windows 7 and Windows 10 environments. The analysis of the results suggests that the proposed model achieved superior level of classification performance for the observed datasets. Additionally, besides conducting rigid statistical tests, best derived model is interpreted by SHapley Additive exPlanations (SHAP) analysis and results findings can be used by security experts to further enhance security of IoT systems.

# INTRODUCTION

During the first two industrial revolutions, production was mechanized, first by the introduction of steam power, then by electrification, and eventually mass production was

made possible by the overall adoption of the production line concept (*Nguyen, 2019*; *Müller, 2019*).

In the late '50s, digital electronics become more common in the predominantly mechanical production, which was the milestone commonly marked as the beginning of the 3rd industrial revolution, *i.e.*, digital revolution. In the following two decades, production lines were automated with the introduction of industrial robots and the digitization of data. This period was also distinguished by the beginning of the widespread use of information technologies (IT), mobile devices and the establishment of a complex communication infrastructure, necessary for further technological progress within the framework of the 4th industrial revolution, also known as Industry 4.0 (*Müller, 2019*; *Veza, Mladineo & Gjeldum, 2015*; *Sani et al., 2020*).

## Industry 4.0

The Domain of Industry 4.0 includes Cyber-Physical Systems (CPS), IoT, Industrial IoT (IIoT), AI, big data, digital twins and other technologies without which contemporary factories could not exist (*Anbesh et al., 2021*). The United Nations' (UN) Sustainability 2030 agenda highlights the production efficiency with minimal use of resources as the core of the future business strategy, including smart production and industrialization with a low impact on the living environment (*Stock & Seliger, 2016*).

In addition to the intensive application of information technologies, the progress of the 4th industrial revolution is backed by recent significant advances in the fields of ML, Computer Vision (CV) and AI, but also manufacturing technologies as well. 3D printing, for example, enables rapid production of necessary components and parts. This progress supported the transition from traditional factories to the smart factory concept (*Jovanovic et al., 2022b*; *Oztemel & Gursev, 2020*). The described technologies have enabled the emergence of flexible production lines based on CPS in various branches of industry. The basic characteristics of such production lines are modularity and interchangeability, which provides mass production capabilities in accordance with the individual needs of the customer (*Zheng et al., 2021*).

At the moment, most of the research in the field of smart factories is focused on the planning and sustainability of production (*Aiello et al., 2020*; *Oztemel & Gursev, 2020*; *Zhang et al., 2022*), as well as on the shortening of the supply chain, partially due to the current geopolitical events in the world (*Anbesh et al., 2021*; *Zheng et al., 2021*; *Asokan et al., 2022*; *Herawati et al., 2021*; *Little & Sylvester, 2022*; *Mandičák et al., 2021*).

## Internet of Things and smart factories

One of the fundamental terms associated with the 4th industrial revolution is the IoT, which essentially represents a set of devices that use a wireless connection for mutual communication, physical quantities reading with various sensors, processing, sharing and storing information *via* the Internet. Nodes within the IoT system can be electronic or embedded devices, as well as physical objects, which communicate with each other and operate without the need for human intervention (*Abu Khurma, Almomani & Aljarah, 2021*).

The concept of networked smart devices has been well-known for a long time, with working prototypes built in the early '80s, *i.e.*, the Coca-Cola vending machine as the first ARPANET-connected appliance was presented in 1982. The term IoT was coined by Peter T. Lewis in 1985. However, it had to wait almost two decades until the appropriate software and hardware technologies (*e.g.*, sensors, actuators, single-board computers, web servers, data storage and processing infrastructure, protocols, wireless networks, *etc.*) emerged on the market (*Madakam, Ramaswamy & Tripathi, 2015*; *Sharma, 2017*; *Du et al., 2022*). Nowadays, the IoT makes possible the collection and exchange of large amounts of data between various sensors, electronic devices, and computer applications, regardless of their mutual distance or geographic location of nodes, which in turn helps in production automation and making timely decisions.

The number of devices within IoT rapidly increases. While in 2015 the total number of devices was 3.6 billion, according to current estimates in 2022 this number is between 11.5 and 14 billion, while by 2025 it is expected to rise somewhere between 16.4 and 25 billion, depending on the source (*Vailshery, 2022*; *Hasan, 2022*).

Vast amounts of generated or aggregated data increase with the number of devices, and must be stored in order to be subsequently available for processing and analytics. Thus, global data storage capacity increased from 2 ZB in 2010, to 100 ZB in 2022, and will rise to estimated 181 ZB in 2025.

## Benefits and challenges

In the context of smart factories, the improvement of product quality and productivity rates are recognized as the main benefits of the 4th industrial revolution, while the improvement of the quality of services is noticeable in health (*Jovanovic et al., 2022b*). The general progress in the field of IT, such as affordable computing resources and emergence of appropriate algorithms, has led to rapid development of technologies, especially AI and ML, which in turn evolved into base tools for solving difficult problems in various fields, including finance, industry, healthcare, human resources, software development and defect prediction, agriculture and logistics, just to name few (*Buchanan, 2019*; *Peres et al., 2020*; *Yu, Beam & Kohane, 2018*; *Dobrojevic & Bacanin, 2022*; *AVSystem, 2020*; *Biliavska, Castanho & Vulevic, 2022*; *Zheng et al., 2022*; *Holliday, Sani & Willett, 2015*; *Muhammad, Abdullah & Samsiah Sani, 2021*; *Mohamed Nafuri et al., 2022*; *Abdul Rahman et al., 2021*).

Healthcare 4.0 represents a critical field of research that is directly related to the development of IoT and ML technologies. During the ongoing COVID-19 pandemic, the technologies of the 4th industrial revolution led to the development of digital solutions that provided tools for efficient management during the crisis of supply chain, human and materials resources, protection of medical personnel, and provided means for remote work of people in wide variety of industries as well (*Javaid et al., 2020*).

Besides direct improvement of conditions and quality of life, diagnostics and prevention play an increasingly important role in providing adequate and timely treatment, especially in diseases such as cancer and diabetes (*Howell, 2010*; *Hopek & Siniak, 2020*).

In scenarios when several medical conditions share similar symptoms, when symptoms appear only in the late stages of the disease, or when it is difficult to recognize symptoms

and provide a diagnosis for whatever other reason (*Gershon-Cohen, Berger & Curcio, 1966*), timely implementation of adequate treatment is of crucial importance. Even nowadays, health system in diagnosis overly rely on the expertise and experience of doctors, in spite of their limited number and current geographical location. Technologies based on IoT, AI, and ML allow medical staff to work with patients remotely, and provide machines with a certain degree of autonomy in diagnosis, thereby reducing the diagnostic time, the risk of misdiagnosis, and in overall, the pressure on medical staff (*Szolovits, 1988*).

In real-world application, currently available computing resources often may prove to be inadequate for large amounts of data processed by AI and ML models, which in turn may affect the overall system performance and reliability. However, increasingly frequent cyber-attacks on IoT systems call into question the credibility of service providers, and consequently threaten business operations and finance.

Cloud Computing (CC) is recognized as technology capable of handling the large amounts of data and traffic generated by IoT systems, but also introduced numerous risks, *e.g.*, inconsistent performance, security risks, latency, and possibility of network breakdowns (*Sabireen & Neelanarayanan, 2021 Li & Geng, 2023*). More recently, technology of Fog Computing (FC) was introduced in order to deal with these issues as an intermediary between the IoT and CC. The key task of the FC is to provide the data generated by the nearby IoT devices. Performing the task locally at the fog node rather than relaying information to the cloud server, FC may deliver services with higher quality and better response time.

In addition, ML algorithms contain a large number of hyperparameters used to control the learning process, whose values often cannot be resolved in an optimal manner and thus essentially affect the speed and quality of the learning process. Traditionally, these parameters are determined through the trial and error, an approach that may be suitable in simpler scenarios, but inapplicable with more complex problems. The latter requires optimization of the methods for hyperparameters determination, which recently brought it under the spotlight of researchers (*Feurer & Hutter, 2019*).

Operating systems, *e.g.*, Windows and Linux, as well as IoT networks, have security vulnerabilities prone to exploitation in order to provide attackers with access to the system and data, and thus IoT systems have been recognized as prime targets for large-scale cyber attacks. Communication between IoT devices can be intercepted and manipulated, and there is always the possibility of one or more devices in the system malfunctioning. Hacking tools are readily available and easy to use, without any specialized skills required to carry on a successful attack (*Louvieris, Clewley & Liu, 2013*). That is why the detection of failures and potential network intrusions in real time is one of the priorities for the secure and stable operation of the system (*Stone-Gross et al., 2009*).

The AI, and especially ML and deep learning (DL) algorithms can be used to overcome such challenges, as tools for error prediction, intrusion detection, diagnostics, *etc.* Effectiveness of the chosen tool in a given situation directly depends on the feature selection (FS), value of hyperparameters and model training conducted in order to achieve the desired result (*Jovanovic et al., 2022b*). Despite the use of advanced security tools such as firewalls, antivirus software, data encryption or biometric verification

(_Al-Jarrah et al., 2016_), cyber attacks continue to affect organizations and businesses, and attackers exploit system vulnerabilities in order to gain access and perform attacks, _e.g._, theft of sensitive information.

In the last decade, a multitude of IT security solutions based on AI have been introduced, Intrusion Detection Systems (IDS) being one of them (_Kareem et al., 2022_). IDSs proved to be effective in protection of IoT systems (_Ashraf et al., 2021_; _Zhou et al., 2020_) due to the possibility of network traffic analysis, distinguishing the legite from malicious traffic, and determining the type of detected attack. There are two basic types of IDSs:

- _Host-based IDSs (HIDSs)_ are programs on the host machine that autonomously monitor system calls and logs, _i.e._, software agents, with the aim of detecting unauthorized activities.
- _Network-based IDSs_ are placed at key positions in the computer network in order to monitor the traffic.

Another obstacle for ML is the variety of attack types and network traffic features, which makes the problem solving more complex (_Aljawarneh, Aldwairi & Yassein, 2018_). Based on their mechanism, IDSs could be further classified into (_Moustafa et al., 2020_; _Kareem et al., 2022_):

- _Signature-based IDSs_ detect potentially illegal activites based on previously known patterns, _i.e._, signatures (_Freeman et al., 2002_; _De La Hoz et al., 2015_). Signature-based HIDS monitor the host by scanning network traffic, logs and memory dumps. Although fast and reliable, they cannot detect previously unknown attacks. They require regular updating of attack pattern definitions aswell, otherwise they will fail to provide detection even for minor changes in the pattern, which is a well known vulnerability often abused by attackers.
- _Anomaly-based IDSs_ compile a profile of the regular system behaviour, which is afterwards used for detection of harmful actions (_Jose et al., 2018_). This is usually achieved by applying of ML and DL models that are trained on previously collected data. Therefore, they can detect new, previously unknown attacks, as well as mutations of known attacks (_Moustafa et al., 2019_); but with the tradeoff of increased processing power required compared to signature-based IDSs (_Jose et al., 2018_; _Moustafa et al., 2019_). Despite the high rates of false positives (FPs), this approach is useful in detection of innovative types of attack.

## Research summary and contributions

The research presented within this article tries to further improve IoT security by verifying the performance of the hybrid CNN and ELM structure to solve the network attacks classification problem on the relatively novel TON_IoT Windows 7 and Windows 10 datasets (_Moustafa et al., 2020_), that are considered as benchmarks for determination of the efficiency of the intrusion detection systems. The elementary lightweight CNN network is used to perform the feature extraction, while the ELM is employed for classification of

the features extracted by the CNN. The CNNs have proven to be very successful in a variety of complex tasks. Beside image classification, they are capable of automatically discovering hidden patterns in data, which other models are not able to do (*Sharma et al., 2021*). Additionally, lightweight CNN is easy and quick to train. At the other hand, other models have much better classification capabilities compared to the CNN's dense layers, and this fact directed proposed research towards replacing CNN's dense layers with traditional ML model.

There are also some other examples in literature where other models were combined with CNNs, where CNNs perform feature extraction, and other models are assigned to execute classification, for example eXtreme gradient boosting (XGBoost) model (*Thongsuwan et al., 2021*; *Khan et al., 2022*; *Niu et al., 2020*), long short term memory model (LSTM) (*Sun et al., 2020*) and support vector machine (SVM) (*Sun et al., 2019*). In this work, the ELM was chosen as it does not require classical training, instead it only requires initialization of weight and bias values. Therefore, the goal of proposed research is to develop as lighter as possible hybrid ML/DL model to deal with this challenging security task. The extensive literature survey has also revealed that this particular CNN and ELM hybrid combination has never been utilized to address the intrusion detection problem.

However, since the ELM's performance at large extent depends on the randomly initialized weights and biases and the number of neurons in the single hidden layer, this research makes use of a modified version of the SCA for tuning of the ELM for this particular issue with the goal of further improving classification performance of the model. Since the problem of determining ELM's weights and biases and number of neurons for specific problem falls into the category of mixed integer continuous non-deterministic polynomial hard (NP-hard) optimization, the choice of employing metaheuristics in this case is logical because they proved to be very efficient NP-hard problem solvers (*Zivkovic et al., 2022a*, *2020*).

Finally, it is also worth pointing out that metaheuristics-based methods can always be enhanced, by modifications or hybridization with other approaches. According to the no free lunch theorem (*Wolpert & Macready, 1997*), algorithm capable of obtaining the best outcomes for every optimization problem does not exists and for each particular task specific algorithm can be introduced.

In accordance to everything stated above, this article offers the following set of contributions:

- An efficient and lightweight hybrid CNN-ELM model is develop to address IoT security challenges;
- An enhanced version of SCA metaheuristics was developed to specifically target the known limitations exhibited by the elementary SCA variant;
- The suggested devised algorithm was utilized to discover the adequate hyper-parameters' values and improve the ELM classification accuracy as a component of the framework designated for the intrusion detection classification;

- The results attained by the proposed structure were compared to other noteworthy metaheuristics, used in the identical experimental framework to classify the network attacks.

The remainder of this manuscript has been assembled as follows. The next section introduces the basics of neural networks and ELM, together with the fundamentals of metaheuristics optimization. Afterwards, elementary SCA has been explained, together with its known flaws, and the modified SCA has been proposed, together with the suggested classification framework. The next section brings forward the experimental setup, experimental outcomes, statistical analysis and model interpretation. Lastly, the final section summarizes the research, hints the future research possibilities and winds up the manuscript.

## PRELIMINARIES AND RELATED WORKS

This sections provides background related to the methods utilized in this research. First, a brief introduction to the artificial neural networks is given, followed by the theoretical background of the utilized ELM model. Finally, a brief overview of metaheuristics optimization is given.

### Artificial neural networks

Artificial neural networks (ANN) are used to solve problems from different domains, which are difficult to solve or cannot be solved using traditional programming techniques. The ANNs can provide quality results in (un)supervised machine learning tasks (*Krogh, 2008*). The ANNs and its types, *e.g.*, CNNs, recurrent neural networks (RNNs), *etc.*, are extensively used in pattern recognition, classification of objects, and prediction. Various forms of ANN are used in traffic for road management (*Olayode et al., 2021*; *Ren et al., 2022*) and autonomous vehicle control (*Zhang, Jing & Xu, 2021*), in civil engineering to predict the fatigue of structural materials (*Bai et al., 2021*), in the military for quantum communication (*Quach, 2021*) and aerial swarms (*Abdelkader et al., 2021*), in agriculture for detection of plants diseases (*Roy & Bhaduri, 2021*) and assessment of soil suitability (*Vincent et al., 2019*), while in medicine they are used for diagnostics (*Esteva et al., 2017*), pandemic related applications (*Adedotun, 2022*) and classification of heart diseases and diabetes, just to name few.

The ANNs may refer to hardware system or software application with architecture influenced by biological neural networks, such as those in the natural brain (*Bhadeshia, 2008*; *Brahme, 2014*). It is based on a set of interconnected points referred to as nodes, emulating neurons, and each connection (edge, or synapse in nature) transmits a signal (a real number) from one neuron to another. The output of each neuron is determined by a non-linear function of the sum of its inputs. Neurons and edges usually have a weight factor that changes during the learning process, and affects the strength of the signal on the connection. Neurons can also have a threshold, passing the signal through only when the aggregate signal exceeds that threshold. Typically, neurons are arranged in layers (*Bre, Gimenez & Fachinotti, 2018*), and different layers can perform different transformations

on input signals. The input signal travels from the input layer (the first layer) to the output layer (the last layer), and may pass through the intermediate layers (hidden layers) multiple times. Each neuron has a local memory in which it remembers the data it processes.

A feedforward neural network (FNN) is an ANN where connections between nodes, *i.e.*, information, always propagate forward (*Zell, 2003*). Single-layer perceptron (SLP) is the simplest form of ANN, having only two layers, input and output, but unfortunately it is not capable of processing efficiently nonlinearly separable patterns (*Hu, 2014*; *Ojha, Abraham & Snášel, 2017*). Multilayer perceptrons (MLPs) overcome these shortcomings by having one or more hidden layers, making them the most popular form of ANN at the moment. Some of the important advantages they possess are robustness, learning capacity, parallel processing and capacity to generalize (*Faris et al., 2016*).

In common speech the process of "capturing" the unknown information is called "learning" or "training" of an ANN. In mathematical context however, to "learn" refers to adjustment of the weight coefficients in order to satisfy the predefined conditions (*Svozil, Kvasnicka & Pospichal, 1997*). The training of the ANN directly affects the quality of the model, and thus making necessary the optimization of the loss function during the learning process (*Duchi, Hazan & Singer, 2011*; *Zeiler, 2012*; *Kingma & Ba, 2015*; *Cheng et al., 2022*). In general, training processes can be classified as:

- *Supervised training.* ANN 'knows' the desired output, so the weight coefficients are adjusted in such a manner that the calculated and desired outputs are as close as possible.
- *Unsupervised training.* The desired output is not known, the system is provided with a group of facts and then left alone to autotune towards a stable state in a limited number of iterations.

During this process, over-fitting can occur, *i.e.*, significant deviations in training and test accuracy, indicating that the network has learned specific data and cannot properly process data outside that range. This problem can be treated by regularization, and some of the suggested approaches are *batch*, *data augmentation*, *dropout*, *drop connect*, *early stopping*, *L1/L2, etc.*

In DL, CNN is a class of ANN. The CNNs mimic the pattern of connectivity between neurons of a biological visual cortex (*Fukushima, 1980*; *Matsugu et al., 2003*), making them an excellent tool for feature extraction, especially suitable in the field of CV. CNNs use relatively light pre-processing because the network "learns" to optimize filters or kernels through ML, compared to traditional algorithms where these filters must be set manually. That very independence from prior knowledge and human intervention in feature extraction is a major advantage.

## Extreme learning machines

The ELM presents an ML approach applied to single-layer FNNs. This approach randomly activates hidden neurons in the network, followed by processing stages that determine the output weights *via* Moore–Penrose generalized inverse. Application of hidden layers and

non-linear transformations converts input values into ELM features space in higher dimensions, which simplifies the original problem (*Jovanovic et al., 2022b*).

If a training set $\aleph = (x_i, t_i)|x_i \in R^d, t_i \in R^m, i = 1, ...., N]$ has $L$ hidden neurons, and an activation function $g(x)$, outputs may be determined as shown in Eq. (1).

$$\sum_{i=1}^{L} \beta_i g(w_i \cdot x_j + b_i) = y_j, j = 1, ..., N \tag{1}$$

where:

- $\beta_i = [\beta_{i1}, ..., \beta_{im}]^T$ are output weights
- $w_i \cdot x_j$ is inner product of $w_i$ and $x_j$
- $w_i = [w_{iq}m, ...., w_{id}]^T$ are input weights of hidden neurons
- $b_i$ is the input bias

Approximation of the standard parameters $\beta_i, i = 1, ...L$ for a simple FNN may be carried out with the Eq. (2)

$$\sum_{i=1}^{L} \beta_i g(w_i \cdot x_j + b_i) = t_j, j = 1, ..., N \tag{2}$$

Furthermore, the parameter $T$ may be calculated with the use of Eq. (3):

$$T = H\beta \tag{3}$$

where $H$ is the hidden layer output matrix shown in Eq. (4)

$$h = \begin{bmatrix} g(w_1 \cdot x_1 + b_1) & ... & g(w_L \cdot x_1 + b_L) \\ \vdots & ... & \vdots \\ g(w_1 \cdot x_N + b_1) & ... & g(w_L \cdot x_N + b_L) \end{bmatrix}_{N \times L} \tag{4}$$

and, $\beta$ and $T$ shown in Eq. (5)

$$\beta = \begin{bmatrix} \beta_i^T \\ \vdots \\ \beta_L^T \end{bmatrix}_{L \times m} \quad \text{and} \quad T = \begin{bmatrix} t_i^T \\ \vdots \\ t_N^T \end{bmatrix}_{N \times m} \tag{5}$$

Output weights $\beta$ are being determined using the minimum norm least-square as shown Eq. (6)

$$\beta = H^\dagger T \tag{6}$$

where $H^\dagger$ represents the generalized Moore–Penrose inverse of $H$.

Initialization of weight and bias variables with random values is essential for the classifier performance, represents a NP-hard challenge, and needs to be optimized for each particular classification problem.

## Metaheuristics optimization

Metaheuristics optimization is a field of AI modelled after examples often found in social groups in nature (*Hu et al., 2021*), and capable of solving complex real-world problems. Swarm intelligence is one of the most prominent groups of metaheuristics, where algorithms mimic the behavior of an individual in a group, *e.g.*, in a colony, flock or herd, in order to solve the targeted problem.

An important feature of metaheuristics algorithms is their ability to deal with complex tasks using both limited computing resources and limited time frames, which cannot be achieved with a traditional mathematical approach. Single execution of the algorithm cannot guarantee desired results due to the inherent randomness, but each subsequent execution increases the chances of finding the true optimum, and thus such algorithms must run through several iterations. Although each algorithm may possess unique properties, the basic components enabling such algorithms to solve NP-hard problems are:

- *Research*. The algorithm covers large areas within the search space, looking for sub-areas potentially containing better solutions.
- *Exploitation*. The algorithm focuses on certain sub-areas, locating the best solution.

Desired results can be achieved only if the adequate balance between research and exploitation was reached, that suits the specific problem. Most algorithms in this area use search agents tuned to work under simple sets of rules, allowing complex behavior to manifest globally.

Agents can be modeled in different ways, and many algorithms with proven good performance found inspiration in natural phenomena such as bee swarms (*Karaboga, 2010*), ant colonies (*Salami, 2009*), whale flocks (*Mirjalili & Lewis, 2016*), wolf packs (*Mirjalili, Mirjalili & Lewis, 2014*) *etc.*

Inspiration can be drawn from abstract ideas as well. The SCA have origins in trigonometry (*Mirjalili, 2016*), the arithmetic optimization algorithm uses simple mathematical formulations (*Abualigah et al., 2021*), as well as the search algorithm modelling user behavior on social networks (*Liang et al., 2006*).

This makes metaheuristics algorithms a popular choice for ANN tuning, *e.g.*, improvement of the fault prediction capability (*Kayarvizhy, Kanmani & Uthariaraj, 2014*), hyper-parameters optimization (*Zhang & Qiu, 2020*); ELM tuning (*Alshamiri, Singh & Surampudi, 2017*), and in general, solving complex real-world problems, such as intrusion detection (*Zivkovic et al., 2022c*), credit card fraud detection (*Jovanovic et al., 2022a*), problems in wireless sensor networks (WSNs) (*Zivkovic et al., 2020*), task scheduling in cloud-based environments (*Bezdan et al., 2021*), optimization of energy and load balance in 5G networks (*Bacanin et al., 2022*), optimization and tuning of ANNs (*Zivkovic et al., 2022a*), as well as the diagnosis of COVID-19 and prognosis of cases (*Zivkovic et al., 2022b*).

---

**Algorithm 1**  **The pseudo-code of the SCA** (*Gabis et al., 2021*).

**Initialize** randomly a set of solutions $X_i(i = 1, 2, ..., n)$

**while** $(t < T)$ **do**

    **Calculate** the objective value for each solution

    **Update** the destination $(P = X)$

    **Update** the random parameters $r_1$, $r_2$, $r_3$ and $r_4$

    **Update** Update the solutions using Eq. (9)

**end while**

**Return** the destination P

---

# DEVELOPED METHOD AND PROPOSED FRAMEWORK

This section first introduces basics of the original SCA metaheuristics, followed by its observed deficiencies and detailes of proposed improved approach. Finally, this section concluded with introduced hybrid ML framework used for classification and solutions' encoding scheme employed by developed metaheuristics.

## Basic sine cosine algorithm

Mathematical model of the SCA is inspired by the trigonometric functions (*Mirjalili, 2016*). The position updating is conducted according to the specified functions, making them prone to oscillations in the region of the optimum, and the return values fall into the $[-1, 1]$ range. During the initialization phase, the algorithm generates multiple solutions as candidates for the best solution given the constraints of the search area and randomized adaptive parameters control the exploration and exploitation phases, Fig. 1 and the pseudo-code in Algorithm 1.

The position updating is performed as follows (*Mirjalili, 2016*):

$$X_i^{t+1} = X_i^t + r_1 \cdot sin(r_2) \cdot |r_3 \cdot P_i^t - X_i^t| \tag{7}$$
$$X_i^{t+1} = X_i^t + r_1 \cdot cos(r_2) \cdot |r_3 \cdot P_i^t - X_i^t| \tag{8}$$

where

- $i$-th and $t$-th are dimensions
- $i + 1$-th is iteration
- $X_{ij}^t$ and $X_{ij}^{t+1}$ denote the positioning for a given solution in the terms of dimension and iteration
- $r_{1-3}$ is generated pseudo-random number
- $P_{ij}^*$ is the position of the target
- $|...|$ represents the absolute value

The equations are combined with the use of control parameter $r_4$:

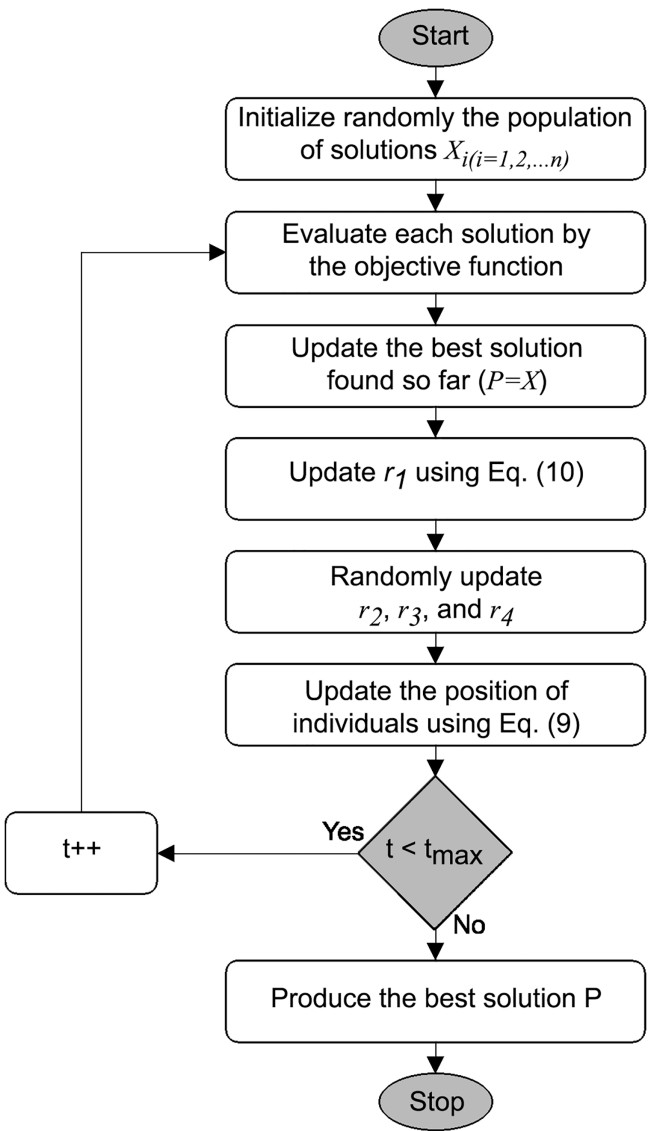

**Figure 1 SCA flowchart, based on *Gabis et al. (2021)*.**

$$X_i^{t+1} = \begin{cases} X_i^{t+1} = X_i^t + r_1 \cdot sin(r_2) \cdot |r_3 \cdot P_i^{*t} - X_i^t|, & r_4 < 0.5 \\ X_i^{t+1} = X_i^t + r_1 \cdot cos(r_2) \cdot |r_3 \cdot P_i^{*t} - X_i^t|, & r_4 \geq 0.5, \end{cases} \tag{9}$$

where $r_4$ denotes a randomly selected number from the $[0, 1]$ interval.

Cyclic sequences due to the sine and cosine functions allow for repositioning near the solution. In order to enhance exploration and the randomness quality, the range for the parameter $r_2$ is set to $[0, 2\Pi]$. The following equation is used to control the diversification and provide the exploitation balance:

$$r_1 = a - t\frac{a}{T}, \tag{10}$$

where:

- $t$ is the current repetition
- $T$ denotes the maximum allowed amount of possible repetitions per run
- $a$ hardcoded, empirically determined value set to 2.0 (as suggested in *Mirjalili (2016)*), not adjustable by the user

The SCA meta-heuristic provides impressive performance with bound-constrained and unconstrained benchmarks, with a relative simplicity and small number of control parameters (*Mirjalili, 2016*). However, when testing with standard Congress on Evolutionary Computation (CEC) benchmarks, the algorithm tends to converge too fast towards current best solutions, with reduced diversity of the population. Due to directed search towards the $P^*$, if the initial results are too far from the optimum, the population will quickly converge towards disadvantageous domain of the search space, with unsatisfactory final results.

## Enhanced sine cosine algorithm

To address the known cons of the basic algorithm, an enhanced version of SCA has been proposed for the sake of the research presented in this article, based on two procedures that have been included in the original metaheuristics:

1) Chaotic initialization of solutions forming the initial population, and
2) Self-adaptive search mechanism switching the search process betwixt classic SCA search and firefly algorithm (FA) search procedure.

The first proposed alternation of the basic version of SCA is chaotic initialization of the starting population. This approach aims to produce the starting set of solutions near the optimum region of the search realm. It was proposed by *Caponetto et al. (2003)*, who embedded the chaotic maps inside metaheuristics algorithms to improve the search phase. Other relevant studies, including *Wang & Chen (2020)*, *Liu et al. (2021)*, *Kose (2018)*, have shown that search procedure efficiency is greater if it relies on chaotic sequences, rather than pseudo-random generators.

There are numerous chaotic maps that can be used, however, empirical experiments executed with SCA metaheuristics have shown that the logistic map yields the most promising results. Consequently, the modified SCA at the begining of the execution utilizes the chaotic sequence $\beta$, starting by the pseudo-random value $\beta_0$, produced by the logistic maping, as given by the Eq. (11).

$$\beta_{i+1} = \mu\beta_i \times (1 - \beta_i), i = 1, 2, \ldots, N - 1, \qquad (11)$$

where $N$ and $\mu$ represent the size of the populace and chaotic control value. The parameter $\mu$ has been set to value 4, while applying the following limitations to $\beta_0$: $0 < \beta_0 < 1$ and $\beta_0 \neq 0.25, 0.5, 0.75, 1$.

Solution $i$ is subjected to mapping according to the produced chaotic sequences for each component $j$ with respect to the following equations:

---

| **Algorithm 2  Pseudo-code of the chaotic-based initialization scheme.** |
| --- |
| Step 1: Produce population *Pop* of $N/2$ individuals by applying conventional initialization mechanism: $X_i = LB + (UB - LB) \cdot rand(0, 1), i = 1, \dots N$, where $rand(0, 1)$ is the pseudo-random value within $[0, 1]$ and *LB* and *UB* are arrays with lower and upper boundaries of every individual's component *j*, respectively. |
| Step 2: Generate the chaotic population $Pop^c$ of $N/2$ solutions by mapping the individuals belonging to *Pop* to chaotic sequences by utilizing Eqs. (11) and (12). |
| Step 3: Integrate *Pop* and $Pop^c$ ($Pop \cup Pop^c$) and sort merged set of size *N* with respect to fitness value in ascending order. |
| Step 4: Ascertain the current best individual *P*. |

$$X_i^c = \beta_i X_i, \tag{12}$$

where the novel location of individual *i* after chaotic disturbances is denoted by $X_i^c$.

The entire chaotic-based generation of the initial population is given in Algorithm 2. It is important to note that the introduced initialization procedure is not affecting the algorithm's complexity with respect to the fitness function evaluations *FFEs*, as it produces only $N/2$ arbitrary individuals, and afterwards it maps those solutions to the corresponding chaotic solutions.

The second alteration of the basic algorithm, the self-adaptive search strategy, is responsible of alternating the search procedure between conventional SCA search and the FA's (*Yang & Slowik, 2020*) search procedure given by Eq. (13).

$$X_i^{t+1} = X_i^t + \beta_0 \cdot e^{-\gamma r_{i,j}^2}(X_j^t - X_i^t) + \alpha^t(\kappa - 0.5), \tag{13}$$

where $\alpha$ denotes the randomization parameter, while $\kappa$ marks the random value drawn from the Gaussian distribution. Distance among two individuals *i* and *j* is represented by $r_{i,j}$. To further improve the FA's exploration and exploitation capabilities, this research utilizes dynamic $\alpha$, as shown in *Yang & Slowik (2020)*.

The proposed SCA algorithm switches between SCA and FA search procedures on the level of every component *j* of every individual *i* as follows: in case that the produced pseudo-random value in range $[0, 1]$ is smaller than the search mode (*sm*), the *j*-th part of individual *i* is updated by applying FA search (Eq. (13), otherwise, conventional SCA search will take place (Eq. (9)). The search mode *sm* control parameter is controlling the balance betwixt SCA and FA search mechanisms, focusing more on the FA search to update the individuals at the beginning. As the iterations go by, assuming that the search realm was explored sufficiently, the SCA search will be activated more often. This is achieved by dynamically reducing the value of *sm* parameter from the starting value in every iteration *t* with respect to:

$$sm_t = sm_{t-1} - \frac{t}{T}. \tag{14}$$

The starting value of *sm* parameter has been established empirically, and set to 0.8 in all simulations executed in this research.

| Algorithm 3    The HASCA pseudo-code. |
| :--- |
| Produce initial population of $N$ individuals by utilizing Algorithm 2. |
| Adjust control parameters and initialize dynamic parameters |
| **while** exit criteria is not reached **do** |
|     **for** each individual $i$ **do** |
|         **for** each component $j$ of individual $i$ **do** |
|             Produce pseudo-random value $rnd$ |
|             **if** $rnd < sm$ **then** |
|                 Update component $j$ by applying FA search (Eq. (13)) |
|             **else** |
|                 Update component $j$ by applying SCA search (Eq. (9)) |
|             **end if** |
|         **end for** |
|     **end for** |
|     Verify the population with respect to the fitness function |
|     Ascertain the current best individual $P$ |
|     Update dynamic parameters' values |
| **end while** |
| Return the best-determined individual |

Finally, the introduced enhanced SCA method is actually a low-level hybrid since the FA search mechanism has been incorporated into the SCA method. It was named hybrid adaptive SCA (HASCA), and the pseudo-code that shows the inner workings of this method is given by Algorithm 3.

At the end, it can be noted that the HASCA is not adding additional overhead to the basic SCA and the complexity with respect to the FFEs of both methods, basic and enhanced, are $O(N) = N \cdot N \cdot T$.

## Proposed classification framework

The suggested classification framework represents a hybrid CNN and ELM structure. The CNN performs the task of feature extraction, where the outputs were collected from the second to the last dense layer (before the final layer). This collection of features were then placed to the inputs of the ELM model that was executing the classification. ELM hyper-parameters were optimized by the proposed HASCA algorithm. The flowchart of this classification framework is shown in Fig. 2.

This suggested framework utilizes an empirically established lightweight CNN structure, with a main purpose to be as elementary as possible to permit effortless training and fast executing. The proposed CNN model is comprised of a singe convolutional layer (64 filters, kernel size 6, with relu activation function), batch normalization, max pooling layer with pool size of three, and a pair of dense coats. Since the experiment was the

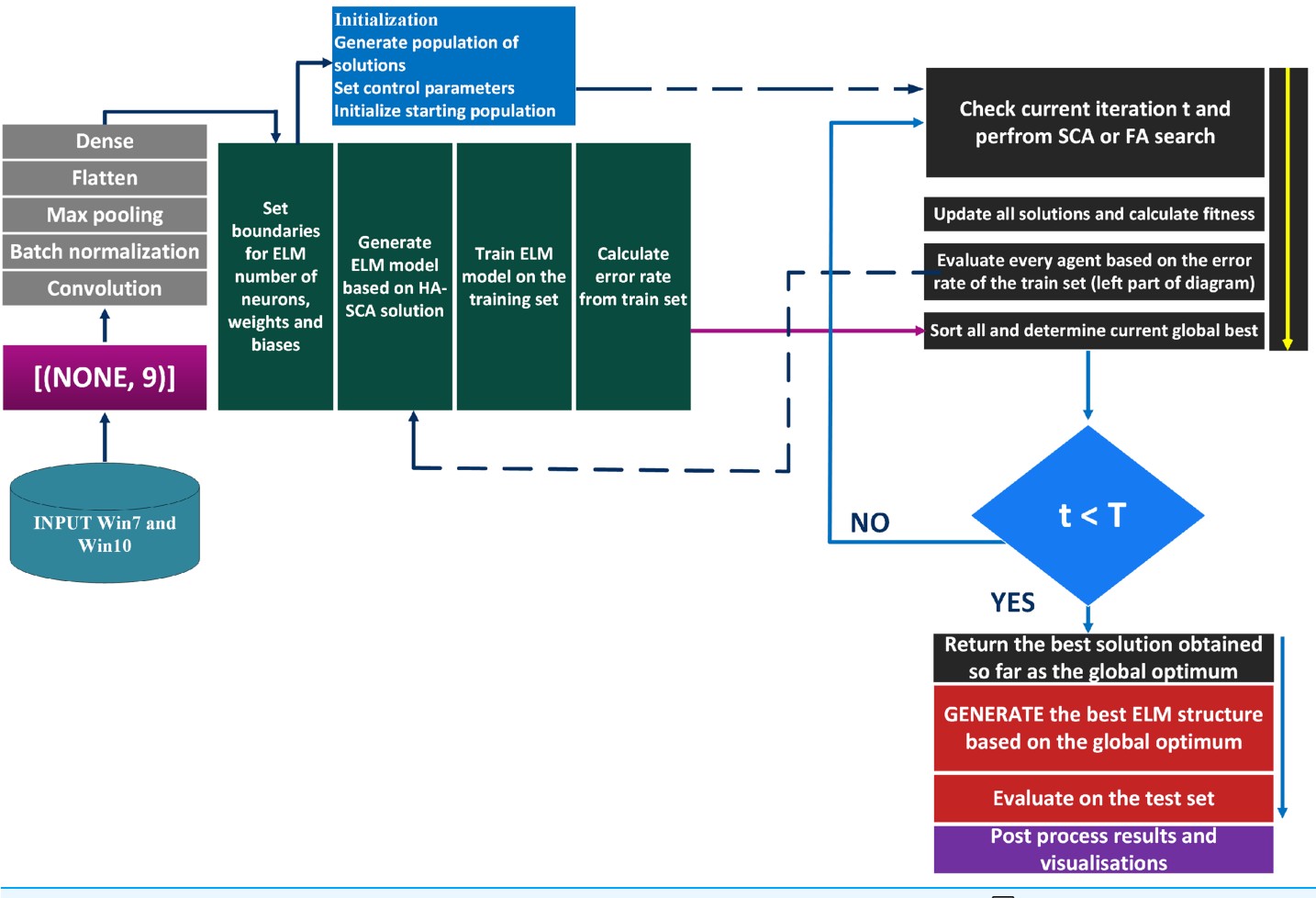

**Figure 2 Flowchart of the suggested classification framework.**

multiclass classifying problem, the common choice for loss is *categorical$_c$rossentropy*, where the adam optimizer was selected with the default learning rate of $lr = 0.001$. To measure the performance level of the model, the accuracy was used. Lastly, this structure has been trained by using the batch size of 16, within 10 epochs. This CNN model is presented in Fig. 3.

## Solutions encoding scheme

The tuning procedure of the ELM structure in this research consisted of optimizing the number of neurons (*nn*) in the hidden layer, together with the weights and biases connecting the input and hidden layers. The bounds for weights and biases were calibrated to the span of $[-1, 1]$, while the limits for the *nn* were calibrated to $[300, 600]$. Here, the *nn* is represented as the integer value, opposite to weights and biases that are continuous inside provided ranges. The optimization of the *nn* corresponds to the ELM hyper-parameters tuning, while the optimization of the weights and biases is related to the training procedure of the ELM model. The range $[300, 600]$ for *nn* has been established

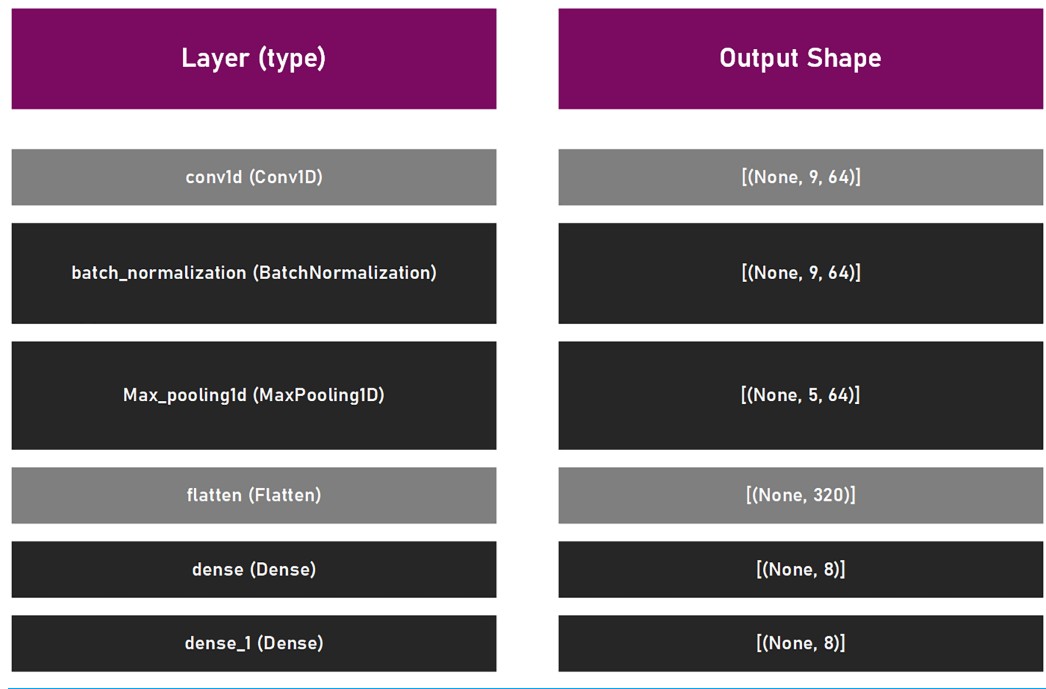

**Figure 3  CNN model that was utilized in suggested framework.**

empirically, aiming to create a network neither too simple or too complex, to avoid overfitting issue. The proposed HASCA algorithm was used to address both described tasks.

Every individual that belongs to the population was encoded by utilizing the regular flat-swarm encoding scheme. In other words, every solution is structured as a vector of length $l$, denoting the count of hyper-parameters that were tuned. As $l$ relies on the count of neural cells $nn$, and length of the input features vector $fs$, it can be obtained as follows: $1 + nn \cdot fs + nn$. More precisely, the first portion of every individual marks the count of neurons (integer), followed by $nn$ elements denoting the biases (continuous), and lastly, the final $nn \cdot fs$ elements denote the weights (continuous).

# EXPERIMENTAL FINDINGS AND DISCUSSION

This section first introduces the datasets utilized throughout the experiments. Afterwards, the experimental setup is briefly explained, followed by the experimental findings and discussion of outcomes. Finally, this section brings forward the statistical tests and validation of the model, together with the SHAP analysis of the most significant features.

## TON_IoT database

In order to properly evaluate the usability of IDS and AI-based security solutions, testing with use of proper datasets gathered in real-world conditions is a must. Various datasets have been proposed in the literature for this purpose (*Koroniotis et al., 2019*; *Moustafa & Slay, 2015*), such as DARPA 98 and KDD-99 (*Lippmann et al., 2000*), which are now considered outdated due to implemented attack scenarios originating back to 1998, then

```
├ Raw_IoT_dataset
├ Raw_Linux_dataset
├ Raw_Network_dataset
└ Raw_Windows_dataset
  ├ Windows_filtered_normal
  │ ├ win7_normal_1.csv
  │ ├ win7_normal_2.csv
  │ ├ win7_normal_3.csv
  │ ├ win10_normal_1.csv
  │ ├ win10_normal_2.csv
  │ ├ win10_normal_3.csv
  │ └ win10_normal_4.csv
  ├ Windows_filtered_normal_attack
  │ ├ Windows_filtered_backdoor_normal
  │ │ ├ win7_backdoor_normal_1.csv
  │ │ └ win7_backdoor_normal_2.csv
  │ ├ Windows_filtered_DDoS_normal
  │ │ ├ win7_DDoS_normal_1.csv
  │ │ └ win10_DDoS_normal_1.csv
  │ ├ Windows_filtered_DoS_normal
  │ ├ Windows_filtered_injection_normal
  │ ├ Windows_filtered_MIMT_normal
  │ ├ Windows_filtered_password_normal
  │ ├ Windows_filtered_runsomware_normal
  │ ├ Windows_filtered_scanning_normal
  │ └ Windows_filtered_XSS_normal
  ├ Windows_original_normal
  └ Windows_original_normal_attack
```

**Figure 4** **TON-Iot database, reduced file structure.**

ADFA-LD (*Creech & Hu, 2013*) and NGIDS-DS (*Haider et al., 2017*). The mentioned datasets are generated on Linux Operating System (OS), while SSENet-2014, AWSCTD and ADFA-WD datasets are suggested for Windows OS machines (*Moustafa et al., 2020*).

TON IoT belongs to the new generation of Industry 4.0 databases. It was created with the aim to compensate for the perceived shortcomings of existing datasets, such as the lack of data on the behavior of memory, hard drives and processors, as well as data related to IoT. It includes federated data sources collected from IoT service telemetry datasets, Windows and Linux OSs datasets, and network traffic datasets. OS datasets are collected from memory, processor, network, process and hard disk audit trails, Fig. 4. Such datasets can be used to test AI-based cyber security solutions, including IDSs, threat intelligence and hunting, privacy protection and digital forensics.

## Datasets

In the experiment, the TON_IoT datasets for Windows 7 and Windows 10 OS were used, containing 133 *vs.* 125 features respectively, and significant amounts of data (28,367 *vs* 35,975 records) generated using Virtual Machines with appropriate OSs and Windows performance tracking tools, as described in *Tableau (2020)*.

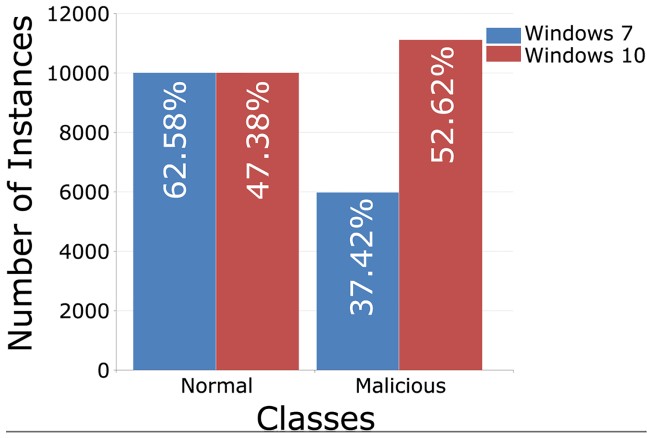

**Figure 5 TON_IoT binary distribution of classes, Windows 7 *vs.* Windows 10.**

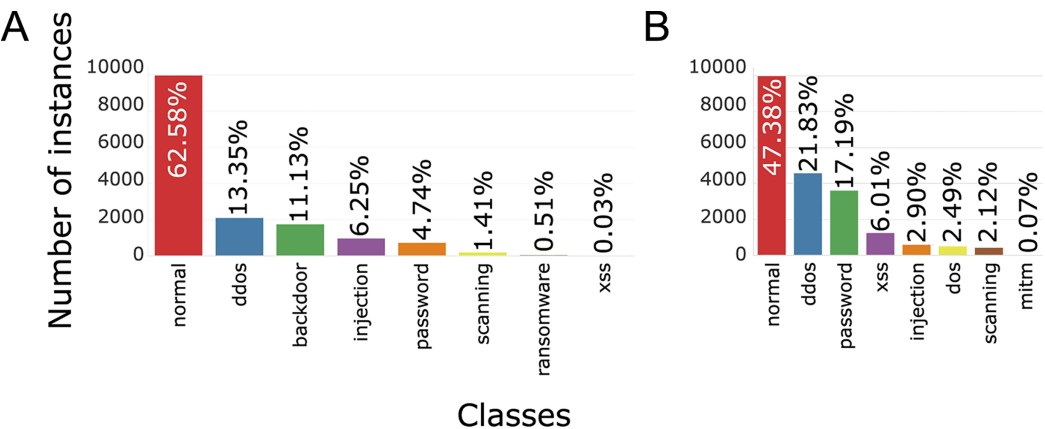

**Figure 6 TON_IoT multiclass distribution of classes, Windows 7 (A) and Windows 10 (B).**

Each training set contains 10,000 records from regular traffic, as well as data classified as attacks, 5,980 for Windows 7 and 11,104 for Windows 10, Figs. 5 (binary distribution) and 6 (multiclass distribution). These sets furthermore can be divided into subsets in the 70%/30% ratio, for training and for testing of the AI model (*Moustafa et al., 2020*).

Correlation analysis shows the importance of features and their use value. A custom correlation function can be used to determine the correlation coefficient between features without a label and to rank the features by strength in the range [−1, 1]. The sign of the correlation coefficient indicates the direction of the connection, while the coefficient itself indicates the strength of connection between two features (*Koroniotis et al., 2019*). A correlation matrix, Figs. 7 and 8, proved useful representing the most closely related features, listed in Tables 1 and 2 which will then be used to train and validate the effectiveness of the ML model in the classification of attacks from the data set. From the presented correlation matrices, it can be noticed that in both datasets exists high positive correlation between a pair of features. In the case of Win 7 benchmark there is 99% correlation between "IO data bytes sec" and "Process (_Total) IO Read Bytes sec"

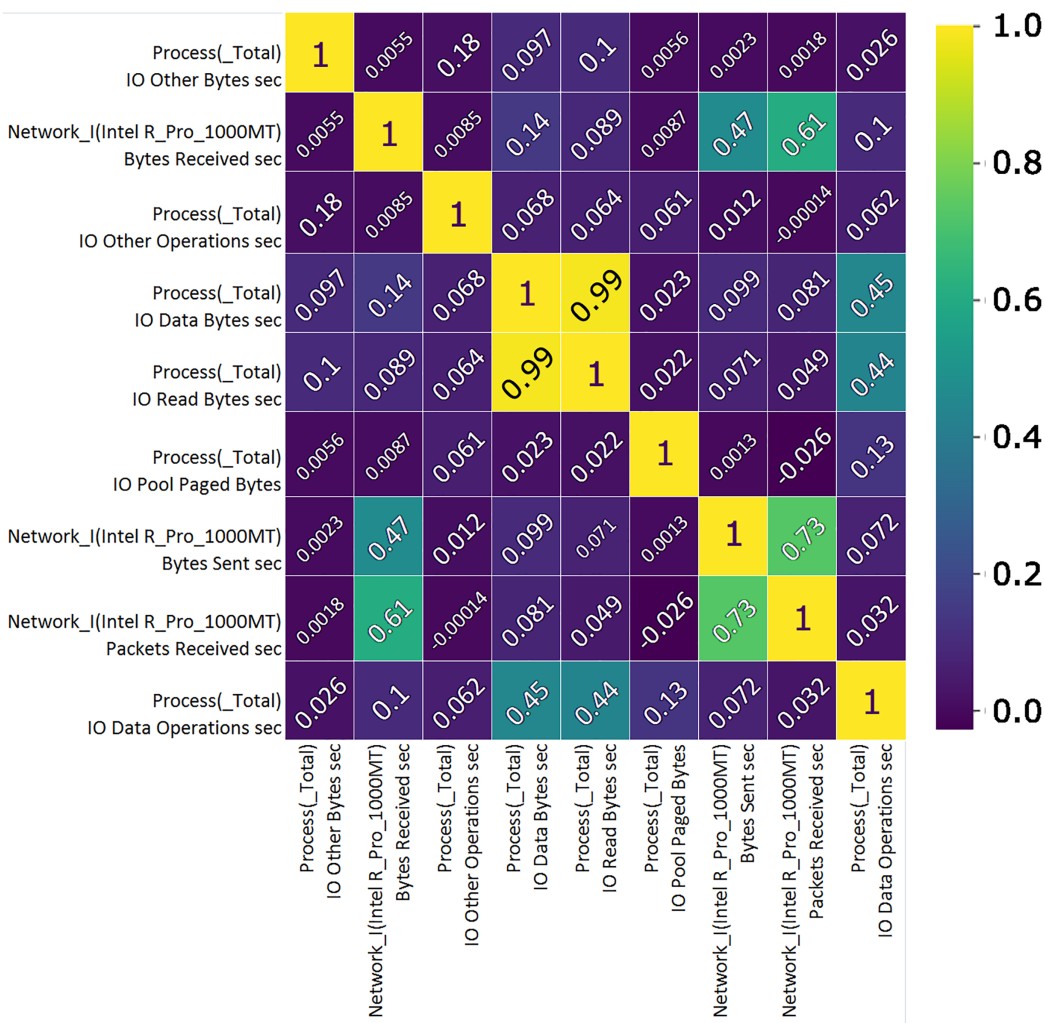

**Figure 7 TON_IoT Windows 7—the most important features correlation matrix.**

attributes, while in the case of Win 10 dataset, there is a correlation of 93% between "Disk ready bytes sec" and "Memory page reads sec" features exists. This seems to be very logical because each pair of features in both datsets refer to the input output (IO) read performance of the system. At the first glance, it may seem as these are redundant (derived) features. However, since the CNN is used for feature extraction in this research, those features were used as input to the model along with other attributes.

## Metrics

The model introduced in this research has been validated according to the conventional machine learning metrics, that rely on true negatives (TN), true positives (TP), false negatives (FN) and FPs projections. These metrics allow obtaining crucial key performance indicators, including accuracy, precision, recall, and F-score.

Moreover, this research utilizes the Cohen's kappa coefficient $\kappa$ (*Cohen, 1960*), that was used as the objective function that is necessary to be maximized. Cohen's kappa coefficient

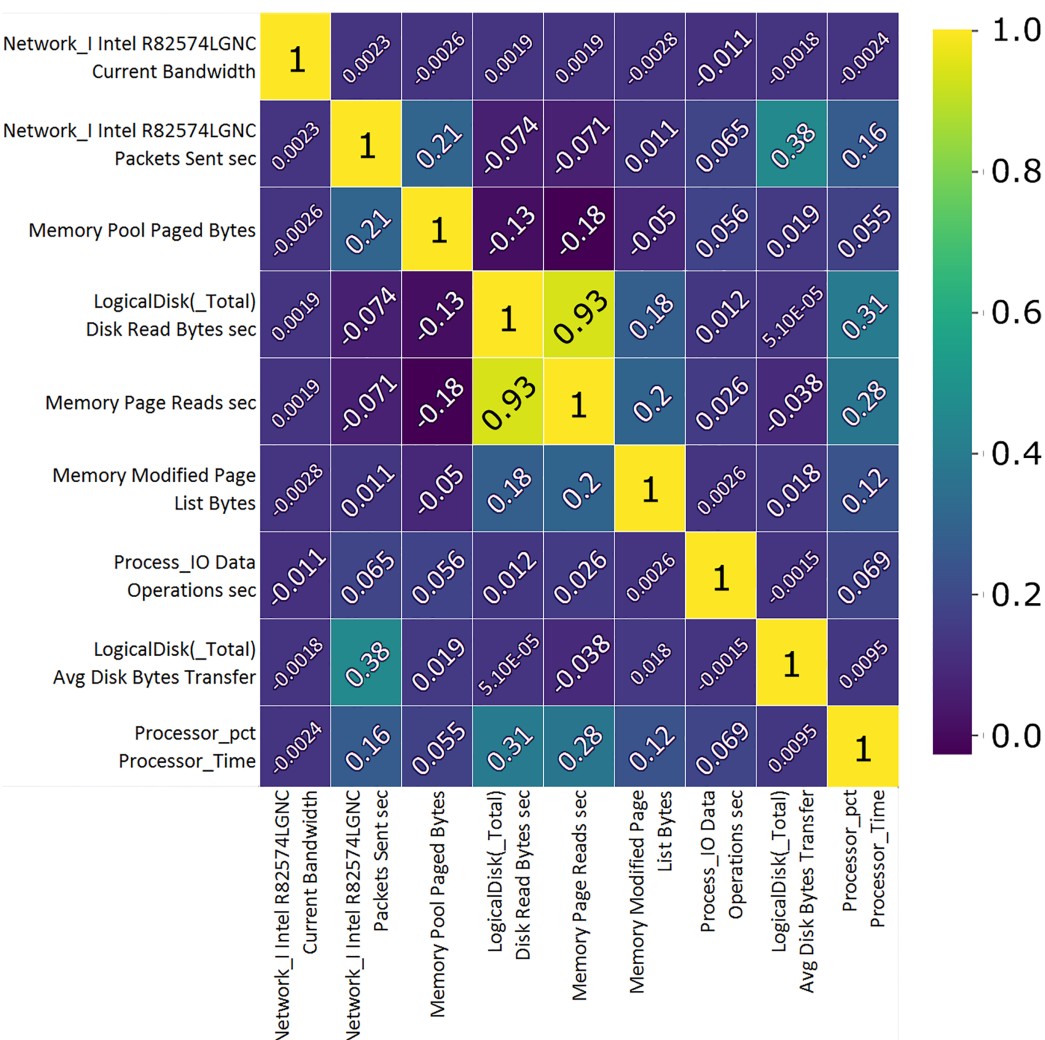

**Figure 8** TON_IoT Windows 10—the most important features correlation matrix.

**Table 1** TON_IoT Windows 7—the most correlated features.

| Feature | Unit | Description |
|---|---|---|
| Proc. total I/O, other | B/s | The rate the bytes are being issued to I/O operations, excluding data related to control operations. |
| Network I.Intel R Pro 1000MT | B/s | Bytes receiving rate over each network adapter, counting framing characters as well. |
| Proc. total I/O, other | OPS | The number of I/O operations issued by process that cannot be classified into read or write operations. |
| Proc. total I/O, data | B/s | The reading or writting rate in I/O operations. |
| Proc. total I/O, read | B/s | The reading rate in I/O operations. |
| Proc. Pool Paged | B | The size of the paged pool in the virtual memory used for objects written to disk when not being used. |
| Network I.Intel.R Pro 1000MT | B/s Sent | Bytes sending rate over each network adapter, counting framing characters as well. |
| Network I.Intel.R Pro 1000MT | PSC | Packets receiving rate on the network interface. |
| Proc. total I/O, data | OPS | The number of I/O operations issued by process, classified as read of write operations. |

**Table 2 TON_IoT Windows 10—the most correlated features.**

| Feature | Unit | Description |
|---|---|---|
| Network I.Intel.R 82574L GNC | B/s | Current network interface bandwidth. |
| Network I.Intel.R 82574L GNC | PSC Sent | Packets transmission rate to subnet-unicast addresses by high level protocols. |
| Memory pool | B | Segment of the paged pool, currently resident and active in physical memory. |
| Logical disk | B/s | Transfer rate from the disk, reading. |
| Memory | B/s | Disk read rate to sort out hard page faults. |
| Memory modified | B | Amount of physical memory assigned to the modified page list. |
| Proc. I/O, data | OPS | The rate of I/O operations, read and write. |
| Logical disk total | B/s | The avg. number of bytes transferred, disk read or write. |
| CPU time | ms | Time required for a non-idle thread. |

denotes a statistical metric that can be utilized to discover inter-rater reliability (*McHugh, 2012*). It is also possible to use it for estimation of the performance level of the given classifier. Cohen's kappa value is calculated from the confusion matrices utilized by machine learning models to evaluate both binary and multiclass classifications. In contrast to the overall accuracy of the classifier, that could be misleading for the imbalanced datasets, Cohen's kappa takes into consideration the imbalance within classes distribution, therefore providing more durable findings. Cohen's kappa is calculated according to the Eq. (15):

$$\kappa = \frac{p_o - p_e}{1 - p_e} = 1 - \frac{1 - p_o}{1 - p_e} \tag{15}$$

where $p_o$ represents the collection of observed values, and $p_e$ denotes the expected values.

## Experimental setup

As previously noted, the ELM model requires tuning for every individual classification task. The developed HASCA algorithm was used to lead the tuning procedure. The outcomes of the ELM optimized by HASCA were put into comparisons to the scores acquired by seven additional powerful metaheuristics, separately implemented for the sake of this research, and deployed in the identical framework as the HASCA, to optimize the ELM model. The chosen contending algorithms were the basic SCA, Artificial bee colony (ABC) (*Karaboga & Basturk, 2008*), bat algorithm (BA) (*Yang & Gandomi, 2012*), whale optimization algorithm (WOA) (*Mirjalili & Lewis, 2016*), elephant herding optimization (EHO) (*Wang, Deb & Coelho, 2015*), chimp optimization algorithm (ChOA) (*Khishe & Mosavi, 2020*) and reptile search algorithm (RSA) (*Abualigah et al., 2022*).

Every competitor metaheuristic has been implemented by making use of the the native control parameter's values as described by its author. Each one of the algorithms was executed with 15 individuals within the population, 15 rounds in each run, and 15 independent executions. In order to simplify the interpretation of the experimental outcomes, prefix CNN-ELM was appended before each metaheuristics (CNN-ELM-

**Table 3 Fitness function results (Cohen's kappa value) over the Win 7 dataset.**

| Method | Best | Worst | Mean | Median | Std | Var | nn |
|---|---|---|---|---|---|---|---|
| CNN-ELM-HASCA | **0.976780** | **0.975701** | **0.976188** | **0.976040** | 3.68E−04 | 1.35E−07 | 552 |
| CNN-ELM-SCA | 0.975291 | 0.974537 | 0.974840 | 0.974892 | **2.72E−04** | **7.39E−08** | 584 |
| CNN-ELM-ABC | 0.974906 | 0.973802 | 0.974260 | 0.974208 | 4.13E−04 | 1.71E−07 | 591 |
| CNN-ELM-BA | 0.974951 | 0.973480 | 0.974506 | 0.974583 | 5.38E−04 | 2.90E−07 | 532 |
| CNN-ELM-WOA | 0.976009 | 0.974202 | 0.975146 | 0.975273 | 5.87E−04 | 3.44E−07 | 596 |
| CNN-ELM-EHO | 0.974957 | 0.974180 | 0.974565 | 0.974537 | 3.33E−04 | 1.11E−07 | 561 |
| CNN-ELM-ChOA | 0.976051 | 0.974211 | 0.975160 | 0.975315 | 6.02E−04 | 3.63E−07 | 567 |
| CNN-ELM-RSA | 0.974931 | 0.974161 | 0.974478 | 0.974208 | 3.68E−04 | 1.35E−07 | 559 |

Note:
The best scores in every considered category are accentuated in bold.

**Table 4 Error metrics scores over the Win 7 dataset.**

| Method | Best | Worst | Mean | Median | Std | Var |
|---|---|---|---|---|---|---|
| CNN-ELM-HASCA | **0.013269** | **0.013901** | **0.013606** | **0.013690** | 2.15E−04 | 4.61E−08 |
| CNN-ELM-SCA | 0.014111 | 0.014532 | 0.014364 | 0.014322 | **1.58E−04** | **2.48E−08** |
| CNN-ELM-ABC | 0.014322 | 0.014954 | 0.014701 | 0.014743 | 2.46E−04 | 6.03E−08 |
| CNN-ELM-BA | 0.014322 | 0.015164 | 0.014575 | 0.014532 | 3.10E−04 | 9.58E−08 |
| CNN-ELM-WOA | 0.013690 | 0.014743 | 0.014195 | 0.014111 | 3.42E−04 | 1.17E−07 |
| CNN-ELM-EHO | 0.014322 | 0.014743 | 0.014532 | 0.014532 | 1.88E−04 | 3.55E−08 |
| CNN-ELM-ChOA | 0.013690 | 0.014743 | 0.014195 | 0.014111 | 3.42E−04 | 1.17E−07 |
| CNN-ELM-RSA | 0.014322 | 0.014743 | 0.014575 | 0.014743 | 2.06E−04 | 4.26E−08 |

Note:
The best scores in every considered category are accentuated in bold.

HASCA represents CNN ELM hybrid model being optimized by the HASCA algorithm, *etc.*).

## Experimental results and discussion

This section presents the simulation outcomes over both considered datasets, and discussed the attained experimental results. For all tables that contain the experimental results, the best scores in every considered category are accentuated in bold.

### Windows 7 dataset experimental results

Table 3 brings forward the experimental results attained by the regarded models over Windows 7 dataset, in terms of the fitness function (Cohen's kappa value) over 15 independent executions. The suggested CNN-ELM-HASCA model attained supreme results, outclassing every other contender model with respect to the best, worst, mean and median values, while CNN-ELM-SCA model acquired the best standard deviation and variance scores. CNN-ELM-ChOA secured the second place, while CNN-ELM-WOA ended up on third position. The last column in Table 3 contains the determined count of neurons $nn$ in ELM. Since the range of search for $nn$ was set to $[300, 600]$, it is clear that almost all models converged to the upper limit. However, since the performance of the

**Table 5 Comprehensive classification scores for Win 7 dataset.**

|  | CNN-ELM-HASCA | CNN-ELM-SCA | CNN-ELM-ABC | CNN-ELM-BA | CNN-ELM-WOA | CNN-ELM-EHO | CNN-ELM-ChOA | CNN-ELM-RSA |
|---|---|---|---|---|---|---|---|---|
| Accuracy (%) | **98.6724** | 98.5892 | 98.5680 | 98.5683 | 98.6309 | 98.5681 | 98.6309 | 98.5681 |
| Precision 0 | 0.989929 | 0.989940 | 0.989729 | 0.990261 | 0.988951 | 0.990590 | **0.990589** | 0.989929 |
| Precision 1 | 0.993773 | 0.992219 | 0.995241 | **0.995319** | 0.995308 | 0.993772 | 0.993771 | 0.995311 |
| Precision 2 | **0.900000** | 0.882353 | 0.842105 | 0.833333 | 0.842105 | 0.789474 | 0.789474 | 0.842105 |
| Precision 3 | 0.986486 | 0.970100 | 0.979866 | 0.976589 | 0.986486 | 0.979866 | 0.979866 | 0.979866 |
| Precision 4 | **0.990329** | 0.988395 | 0.982692 | 0.986538 | 0.986564 | 0.986513 | 0.986513 | 0.982692 |
| Precision 5 | 1.000000 | 1.000000 | 1.000000 | 1.000000 | 1.000000 | 1.000000 | 1.000000 | 1.000000 |
| Precision 6 | 0.982609 | 0.982609 | 0.982609 | 0.982609 | 0.982609 | 0.982609 | 0.982609 | 0.982609 |
| Precision 7 | 0.762712 | 0.796296 | 0.771930 | 0.745763 | 0.788462 | 0.750000 | **0.793103** | 0.771930 |
| M.Avg. Precision | **0.950729** | 0.950239 | 0.943055 | 0.938801 | 0.946310 | 0.934102 | 0.939491 | 0.943055 |
| W.Avg. Precision | **0.986309** | 0.985270 | 0.985130 | 0.985139 | 0.985591 | 0.985200 | 0.985809 | 0.985131 |
| Recall 0 | 0.994271 | 0.994939 | 0.994611 | 0.993587 | **0.995279** | 0.993589 | 0.994271 | 0.994612 |
| Recall 1 | 1.000000 | 1.000000 | 0.996865 | 1.000000 | 0.998433 | 1.000000 | 1.000000 | 0.996865 |
| Recall 2 | **0.818182** | 0.681818 | 0.727273 | 0.681818 | 0.727273 | 0.681818 | 0.681818 | 0.727273 |
| Recall 3 | 0.986486 | 0.986486 | 0.986486 | 0.986486 | 0.986486 | 0.986486 | 0.986486 | 0.986486 |
| Recall 4 | 0.973384 | 0.971483 | 0.971483 | 0.975285 | **0.977186** | 0.973384 | 0.973384 | 0.971483 |
| Recall 5 | 0.833333 | 0.833333 | 0.833333 | 0.833333 | 0.833333 | 0.833333 | 0.833333 | 0.833333 |
| Recall 6 | 0.995595 | 0.995595 | 0.995595 | 0.995595 | 0.995595 | 0.995595 | 0.995595 | 0.995595 |
| Recall 7 | 0.671642 | 0.641791 | 0.656716 | 0.656716 | 0.611940 | 0.671642 | **0.686567** | 0.656716 |
| M.Avg. Recall | **0.909111** | 0.888181 | 0.895295 | 0.890354 | 0.890691 | 0.891982 | 0.893932 | 0.895295 |
| W.Avg. Recall | **0.986729** | 0.985890 | 0.985681 | 0.985681 | 0.986309 | 0.985681 | 0.986309 | 0.985681 |
| F1-score 0 | 0.992088 | **0.992420** | 0.992259 | 0.991919 | 0.992099 | 0.992094 | 0.992431 | 0.992259 |
| F1-score 1 | 0.996881 | 0.996089 | 0.996090 | **0.997648** | 0.996869 | 0.996869 | 0.996869 | 0.996080 |
| F1-score 2 | **0.857138** | 0.769228 | 0.780491 | 0.750012 | 0.780490 | 0.731711 | 0.731714 | 0.780491 |
| F1-score 3 | 0.986492 | 0.978217 | 0.983173 | 0.981509 | 0.986491 | 0.983170 | 0.983159 | 0.983173 |
| F1-score 4 | 0.981772 | 0.979872 | 0.977049 | 0.980879 | **0.981848** | 0.979899 | 0.979912 | 0.977049 |
| F1-score 5 | 0.909087 | 0.909089 | 0.909089 | 0.909086 | 0.909089 | 0.909087 | 0.909086 | 0.909089 |
| F1-score 6 | 0.989062 | 0.989060 | 0.989061 | 0.989062 | 0.989060 | 0.989063 | 0.989062 | 0.989064 |
| F1-score 7 | 0.714290 | 0.710738 | 0.709683 | 0.698409 | 0.689081 | 0.708657 | **0.736012** | 0.709669 |
| M.Avg. F1-score | **0.928349** | 0.915587 | 0.91709 | 0.912320 | 0.915632 | 0.911321 | 0.914781 | 0.917109 |
| W.Avg. F1-score | **0.986449** | 0.985372 | 0.985298 | 0.985316 | 0.985750 | 0.985370 | 0.985968 | 0.985298 |

**Note:**
The best scores in every considered category are accentuated in bold.

model significantly depends of the weight and biases, the proposed CNN-ELM-HASCA attains the best results with just 552 neurons, than CNN-ELM-BA (532 neurons), or other models that generated structures with more neurons.

Regarding the classifying error ratio on Windows 7 dataset, Table 4 yields the scores for every contending model. Similarly to the fitness function scores, the proposed CNN-ELM-

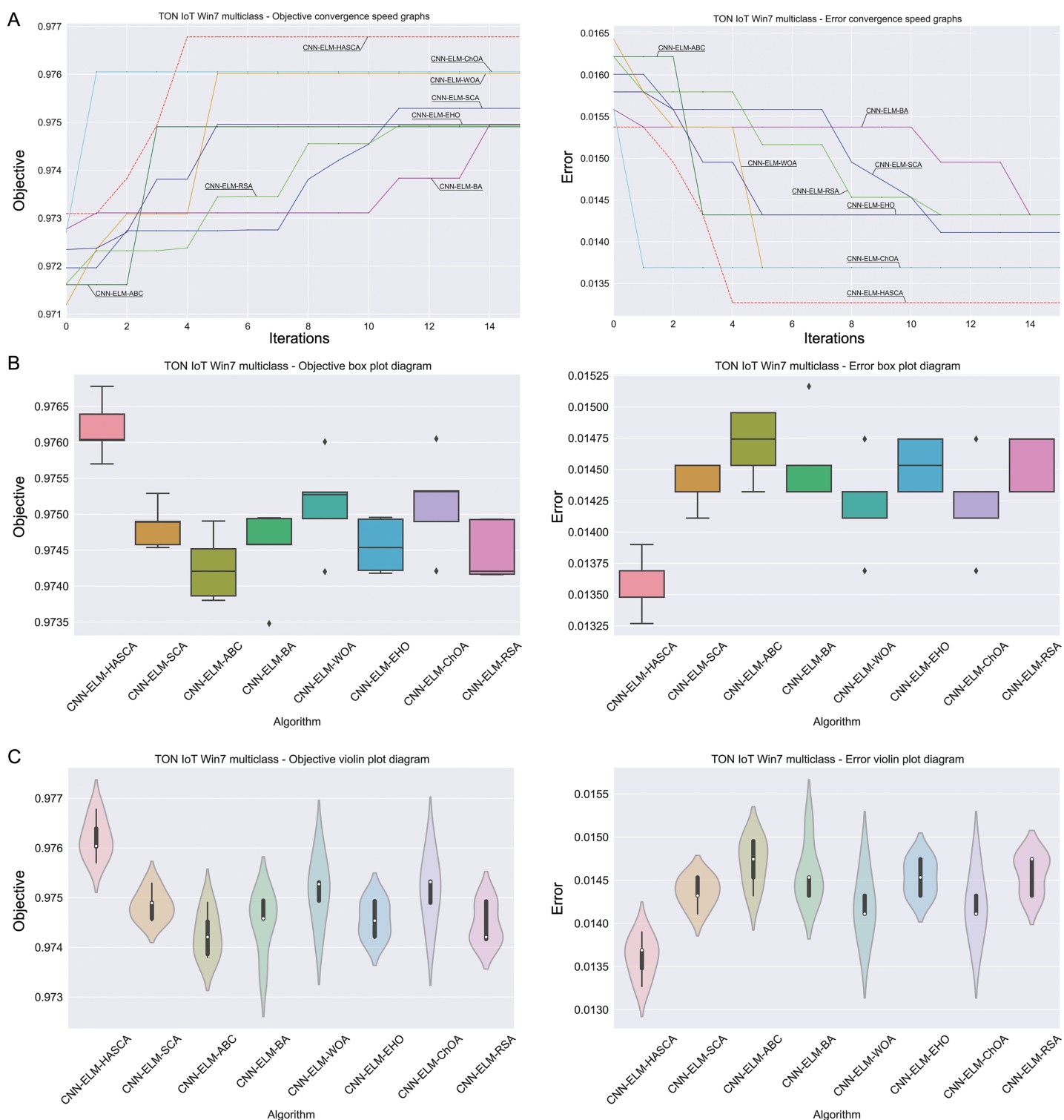

**Figure 9** Visual representations of the simulation outcomes in terms of convergence (A), box plots (B), and violin plots (C) with respect to the objective function (Cohen's kappa) and classification error rate on Win 7 dataset.

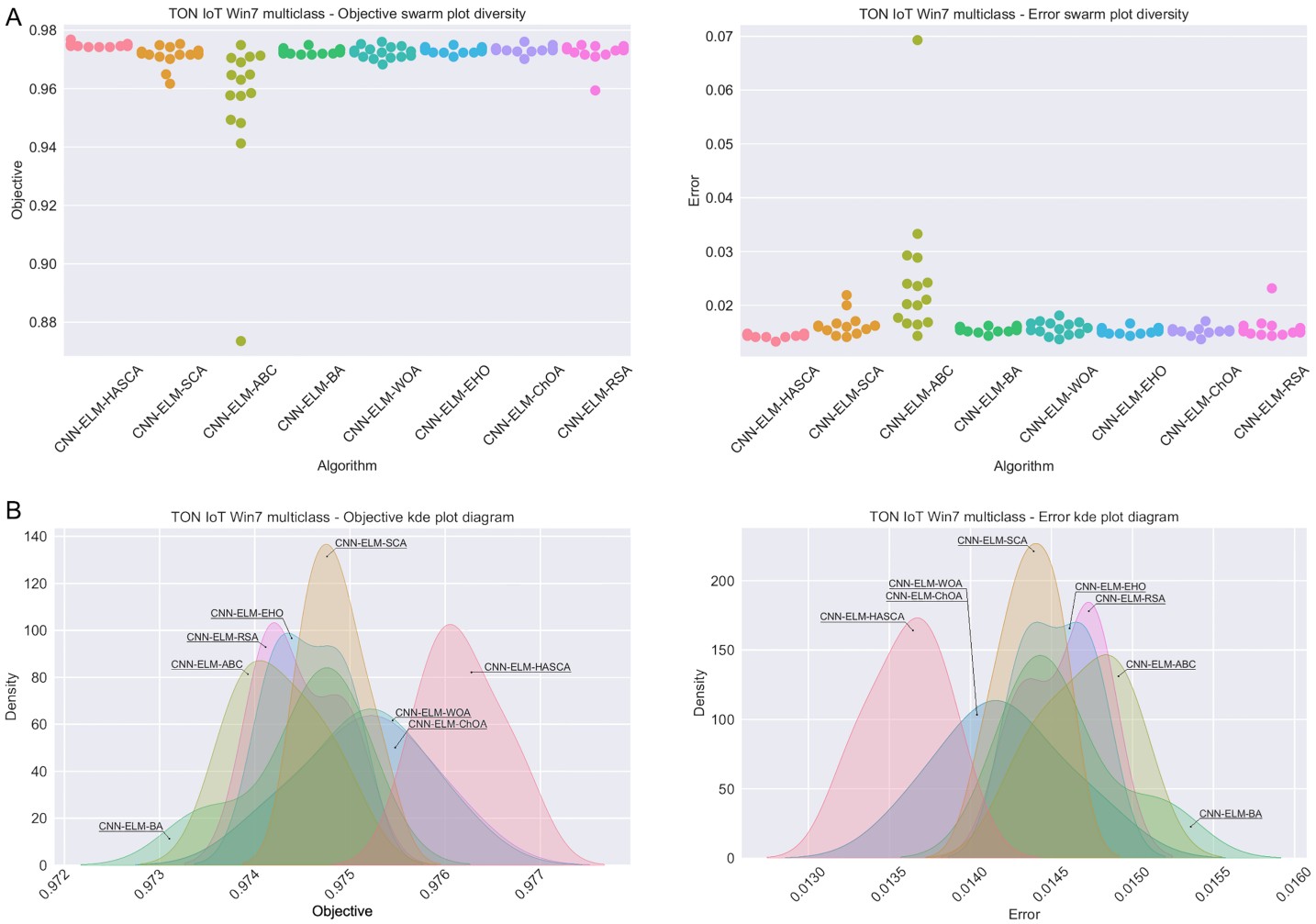

**Figure 10** Visual representations of the simulation outcomes in terms of swarm plot diagrams (A) and KDE plot diagrams (B) related to the objective function (Cohen's kappa) and classification error rate on Win 7 dataset.

HASCA model once more outclassed every other contender, with respect to the best, worst, mean and median values, leaving behind CNN-ELM-ChOA and CNN-ELM-WOA approaches. Concerning standard deviation and variance values, CNN-ELM-SCA again attained the best results, providing most stable results over the runs.

Table 5 brings forward the comprehensive metrics achieved by the best run of every regarded model. It is worth noting that the suggested CNN-ELM-HASCA model attained the superior accuracy level of 98.67%, while CNN-ELM-ChOA and CNN-ELM-WOA acquired the second best accuracy of 98.63%. The suggested CNN-ELM-HASCA method has also displayed supremacy when taking into account other statistical categories, as it attained the best results for the majority of other indicators.

Visualisations of the experimental outcomes attained on the Windows 7 dataset have been provided in Figs. 9 and 10. Figure 9 exposes the convergence graphs, box and violin
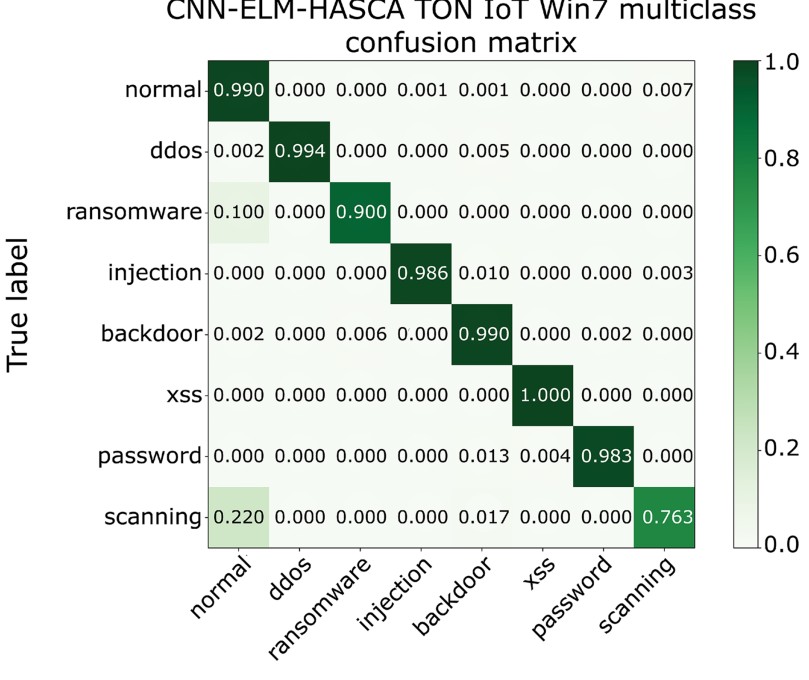

**Figure 11** **The confusion matrix of the proposed CNN-ELM-HASCA method on Win 7 dataset.**

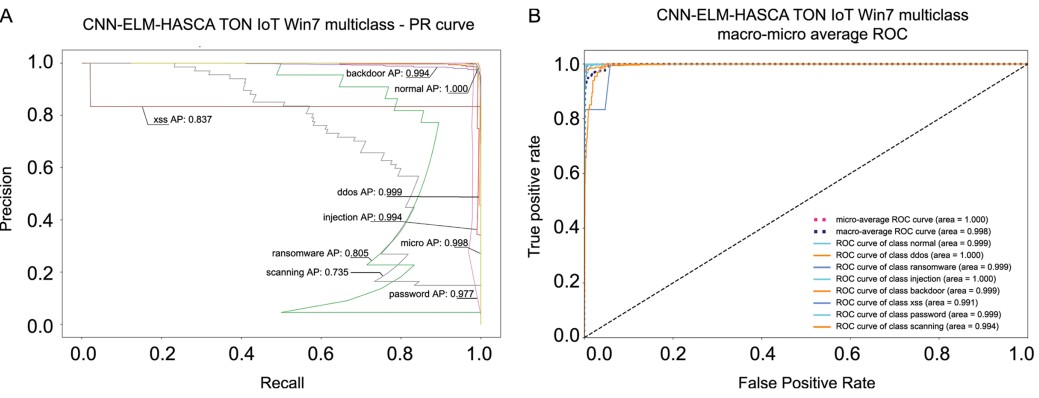

**Figure 12** **Visual representations of the simulation outcomes in terms of PR AUC (A) and ROC AUC (B) curves of the proposed CNN-ELM-HASCA method on Win 7 dataset.**

plots of all noted algorithms with respect to the fitness function (Cohen's kappa in experiments) and classifying error rate. The convergence graphs exhibit the supremacy of the HASCA method, as it is possible to note that employed switching mechanism betwixt SCA and FA search processes significantly aids to the fast converging capabilities of the suggested approach. Additionally, Fig. 10 exhibits the swarm plots making possible to estimate the diversity of solutions during the last iteration of the best run for every noted method, with respect to the fitness function and classification error rate. Once more, one

**Table 6 Fitness function results (Cohen's kappa value) over the Win 10 dataset.**

| Method | Best | Worst | Mean | Median | Std | Var | *nn* |
|---|---|---|---|---|---|---|---|
| CNN-ELM-HASCA | **0.951796** | **0.951079** | **0.951411** | **0.951343** | **2.37E−04** | **5.61E−08** | 539 |
| CNN-ELM-SCA | 0.951311 | 0.950179 | 0.950827 | 0.950880 | 3.93E−04 | 1.54E−07 | 599 |
| CNN-ELM-ABC | 0.949262 | 0.948551 | 0.949021 | 0.949037 | 2.59E−04 | 6.71E−08 | 561 |
| CNN-ELM-BA | 0.950428 | 0.948315 | 0.949410 | 0.949298 | 7.35E−04 | 5.40E−07 | 544 |
| CNN-ELM-WOA | 0.950873 | 0.948776 | 0.949947 | 0.950408 | 7.93E−04 | 6.28E−07 | 536 |
| CNN-ELM-EHO | 0.949951 | 0.947016 | 0.948506 | 0.948829 | 1.18E−03 | 1.40E−06 | 584 |
| CNN-ELM-ChOA | 0.951567 | 0.949507 | 0.950644 | 0.950864 | 6.77E−04 | 4.59E−07 | 532 |
| CNN-ELM-RSA | 0.950410 | 0.949265 | 0.949866 | 0.949965 | 3.74E−04 | 1.40E−07 | 587 |

Note:
The best scores in every considered category are accentuated in bold.

**Table 7 Error metrics scores over the Win 10 dataset.**

| Method | Best | Worst | Mean | Median | Std | Var |
|---|---|---|---|---|---|---|
| CNN-ELM-HASCA | **0.033487** | **0.033963** | **0.033741** | **0.033804** | **1.62E−04** | **2.62E−08** |
| CNN-ELM-SCA | 0.033804 | 0.034598 | 0.034153 | 0.034122 | 2.73E−04 | 7.46E−08 |
| CNN-ELM-ABC | 0.035233 | 0.035709 | 0.035391 | 0.035391 | 1.74E−04 | 3.02E−08 |
| CNN-ELM-BA | 0.034439 | 0.035867 | 0.035137 | 0.035233 | 4.98E−04 | 2.48E−07 |
| CNN-ELM-WOA | 0.034122 | 0.035550 | 0.034756 | 0.034439 | 5.41E−04 | 2.92E−07 |
| CNN-ELM-EHO | 0.034756 | 0.036820 | 0.035772 | 0.035550 | 8.31E−04 | 6.91E−07 |
| CNN-ELM-ChOA | 0.033645 | 0.035074 | 0.034280 | 0.034122 | 4.71E−04 | 2.22E−07 |
| CNN-ELM-RSA | 0.034439 | 0.035233 | 0.034820 | 0.034756 | 2.58E−04 | 6.65E−08 |

Note:
The best scores in every considered category are accentuated in bold.

can take notice that each one of the solutions during the last iteration of HASCA run have been closely placed in the proximity of the best individual. Lastly, Fig. 10 also provides kernel density estimation diagrams (KDE), used to visually show that the simulation results belong to the normal distribution.

Classification algorithm's performance level is commonly defined by utilizing a confusion matrix, that visually presents the classification accuracy and errors. Moreover, the precision-recall (PR) and receiver operating characteristics (ROC) curves are two very important visual tools for classification tasks. Area under PR curve (PR AUC) is combining the precision and recall within single plot, while area under ROC curve (ROC AUC) shows the trade-off betwixt the true positive and false positive rates. Therefore, to further visualize the performance of the proposed CNN-ELM-HASCA approach on Win 7 dataset, Fig. 11 displays the confusion matrix, while Fig. 12 depicts PR AUC and ROC AUC curves achieved by the suggested method.

### Windows 10 dataset experimental results

Table 6 displays the simulation outcomes attained by the regarded models over Windows 10 dataset, with respect to the fitness function (Cohen's kappa score) over 15 independent executions. The suggested CNN-ELM-HASCA model once again attained supreme results,

**Table 8 Comprehensive classification scores for Win 10 dataset.**

| | CNN-ELM-HASCA | CNN-ELM-SCA | CNN-ELM-ABC | CNN-ELM-BA | CNN-ELM-WOA | CNN-ELM-EHO | CNN-ELM-ChOA | CNN-ELM-RSA |
|---|---|---|---|---|---|---|---|---|
| Accuracy (%) | **96.6509** | 96.6189 | 96.4771 | 96.5559 | 96.5881 | 96.5239 | 96.6360 | 96.5559 |
| Precision 0 | **0.979871** | 0.978571 | 0.978223 | 0.978519 | 0.979221 | 0.979211 | 0.979211 | 0.978879 |
| Precision 1 | **0.924662** | 0.918371 | 0.912159 | 0.912158 | 0.918372 | 0.912159 | 0.917812 | 0.918371 |
| Precision 2 | 0.980491 | 0.981884 | 0.976879 | 0.980463 | 0.979725 | 0.976208 | **0.983309** | 0.981884 |
| Precision 3 | 0.846154 | **0.867816** | 0.853107 | 0.862857 | 0.850829 | 0.858757 | 0.853933 | 0.853933 |
| Precision 4 | 0.935310 | **0.937838** | 0.935484 | 0.935484 | 0.932796 | 0.927614 | 0.935484 | 0.934783 |
| Precision 5 | 0.971455 | 0.968836 | **0.974217** | 0.971481 | 0.972376 | 0.973297 | 0.971507 | 0.968864 |
| Precision 6 | **0.758065** | 0.744000 | 0.721311 | 0.736434 | 0.747967 | 0.750000 | 0.736434 | 0.736000 |
| Precision 7 | 1.000000 | 1.000000 | 1.000000 | 1.000000 | 1.000000 | 1.000000 | 1.000000 | 1.000000 |
| M.Avg. Precision | 0.924500 | **0.924663** | 0.918922 | 0.922176 | 0.922659 | 0.922156 | 0.922210 | 0.921589 |
| W.Avg. Precision | **0.966079** | 0.96571 | 0.964119 | 0.965181 | 0.965379 | 0.964579 | 0.966011 | 0.965050 |
| Recall 0 | 0.985818 | **0.986489** | 0.985478 | 0.984469 | 0.986161 | 0.985819 | 0.985819 | 0.985819 |
| Recall 1 | 0.870971 | 0.870971 | 0.870971 | 0.870970 | 0.870971 | 0.870970 | 0.864521 | 0.870971 |
| Recall 2 | **0.986909** | 0.985455 | 0.983273 | 0.985455 | 0.984000 | 0.984727 | 0.985455 | 0.985455 |
| Recall 3 | 0.841530 | 0.825137 | 0.825137 | 0.825137 | 0.841530 | 0.830601 | 0.830601 | 0.830601 |
| Recall 4 | 0.925333 | 0.925333 | 0.92800 | 0.92800 | 0.925333 | 0.922667 | 0.928000 | 0.917333 |
| Recall 5 | 0.971455 | 0.973297 | 0.974217 | 0.972376 | 0.972376 | 0.973297 | 0.973297 | 0.974217 |
| Recall 6 | 0.706767 | 0.699248 | 0.661654 | 0.714286 | 0.691729 | 0.676692 | 0.714286 | 0.691729 |
| Recall 7 | 0.875000 | 0.875000 | 0.875000 | 0.875000 | 0.875000 | 0.875000 | 0.875000 | 0.875000 |
| M.Avg. Recall | **0.895473** | 0.892617 | 0.887966 | 0.894461 | 0.893387 | 0.889971 | 0.894622 | 0.891390 |
| W.Avg. Recall | **0.966509** | 0.966201 | 0.964771 | 0.965559 | 0.965881 | 0.965244 | 0.966349 | 0.965559 |
| F1-score 0 | **0.982829** | 0.982509 | 0.981841 | 0.981491 | 0.982680 | 0.982510 | 0.982510 | 0.982341 |
| F1-score 1 | **0.897009** | 0.894039 | 0.891091 | 0.891091 | 0.894039 | 0.891091 | 0.890371 | 0.894039 |
| F1-score 2 | 0.983689 | 0.983671 | 0.980059 | 0.982949 | 0.981860 | 0.980451 | **0.984379** | 0.983671 |
| F1-score 3 | 0.843841 | 0.845941 | 0.838891 | 0.843580 | **0.846149** | 0.844439 | 0.842110 | 0.842112 |
| F1-score 4 | 0.930288 | 0.931539 | 0.931731 | 0.931730 | 0.929049 | 0.925128 | 0.931731 | 0.925983 |
| F1-score 5 | 0.971447 | 0.971059 | **0.974221** | 0.971931 | 0.972380 | 0.973289 | 0.972399 | 0.971527 |
| F1-score 6 | **0.731521** | 0.720929 | 0.690201 | 0.725189 | 0.718749 | 0.711459 | 0.725189 | 0.713180 |
| F1-score 7 | 0.933329 | 0.933328 | 0.933329 | 0.933329 | 0.933329 | 0.933328 | 0.933327 | 0.933329 |
| M.Avg. F1-score | **0.909251** | 0.907881 | 0.902671 | 0.907659 | 0.907281 | 0.905209 | 0.907749 | 0.905767 |
| W.Avg. F1-score | **0.966228** | 0.965844 | 0.964368 | 0.965315 | 0.965559 | 0.964825 | 0.966125 | 0.965236 |

**Note:**
The best scores in every considered category are accentuated in bold.

outclassing every other contender model with respect to all observed metrics—the best, worst, mean and median values, and also standard deviation and variance scores. Simply said, the suggested model in this case not only acquired the best results, it also put consistent performance level over the independent executions, constantly providing results near to the mean value. CNN-ELM-ChOA secured the second place, while

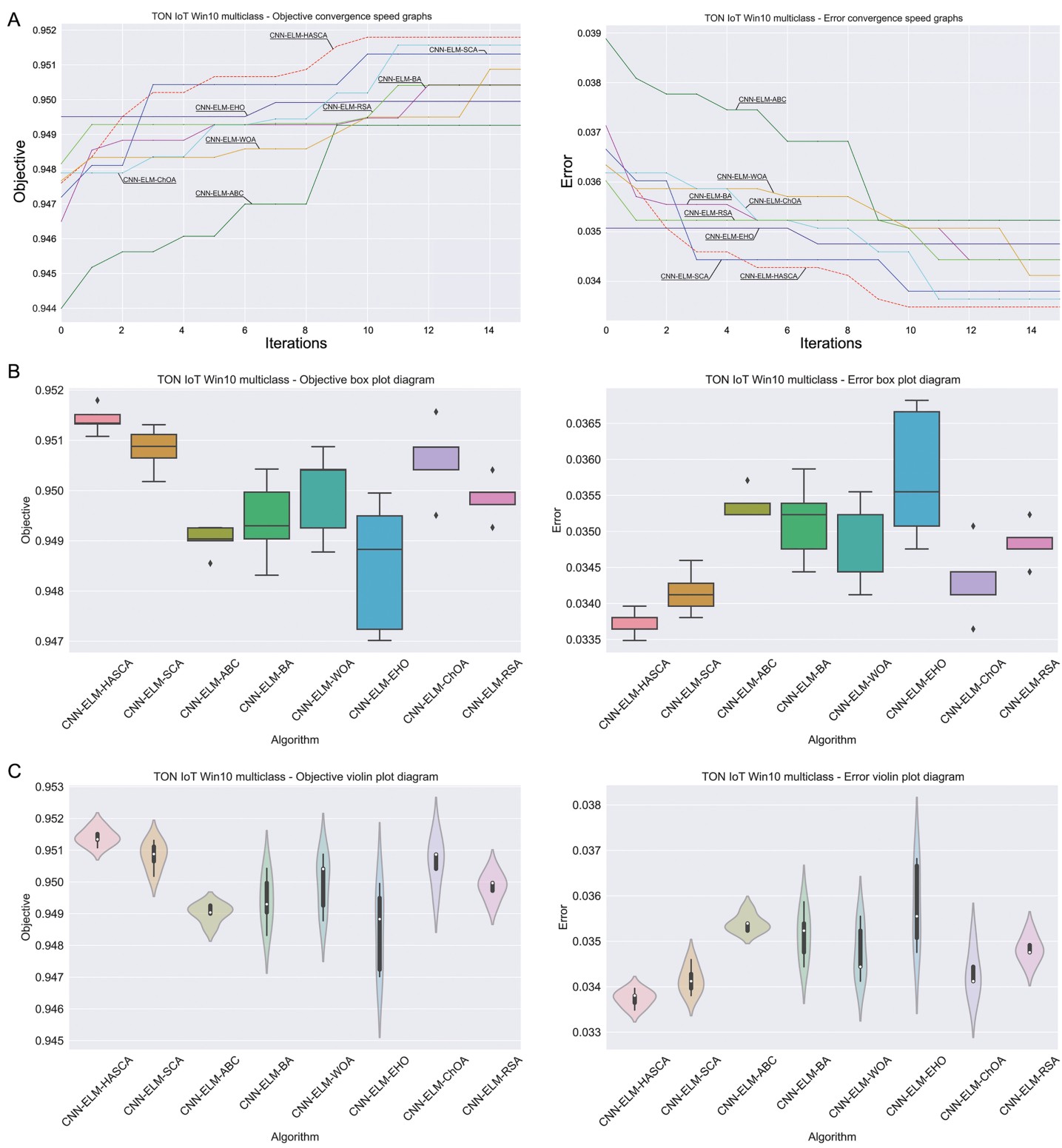

**Figure 13** Visual representations of the simulation outcomes in terms of convergence (A), box plots (B), and violin plots (C) with respect to the objective function (Cohen's kappa) and classification error rate on Win 10 dataset.

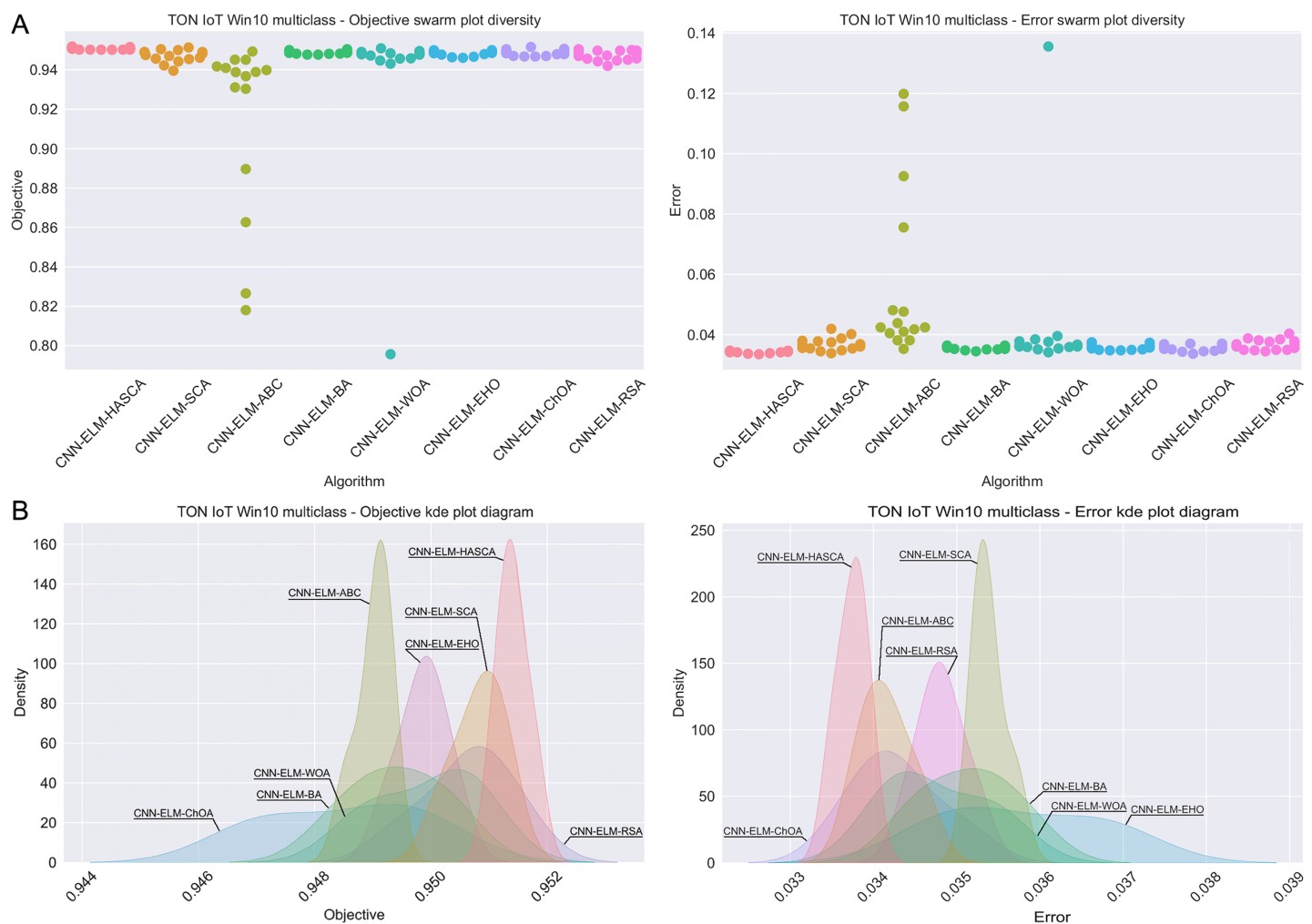

**Figure 14** Visual representations of the simulation outcomes in terms of swarm plot diagrams (A) and KDE plot diagrams (B) related to the objective function (Cohen's kappa) and classification error rate on Win 10 dataset.

CNN-ELM-SCA ended up on third position in this scenario. Similarly to Windows 7 experiments, the last column in Table 6 contains the determined count of neurons *nn* in ELM. Since the range of search for *nn* was set to [300, 600], it is once again obvious that almost all models converged to the upper limit. However, since the performance of the model significantly depends of the weight and biases, the proposed CNN-ELM-HASCA attains the best results with just 539 neurons, than CNN-ELM-WOA and CNN-ELM-ChOA (536 and 532 neurons, respectively), or other algorithms that produced models with more neurons.

With regard to the classifying error ratio achieved on Windows 10 dataset, Table 7 summarizes the scores for each one of the contending models. Similarly to the fitness function scores, the proposed CNN-ELM-HASCA model once more outclassed every other contender, with respect to all observed metrics—the best, worst, mean and median

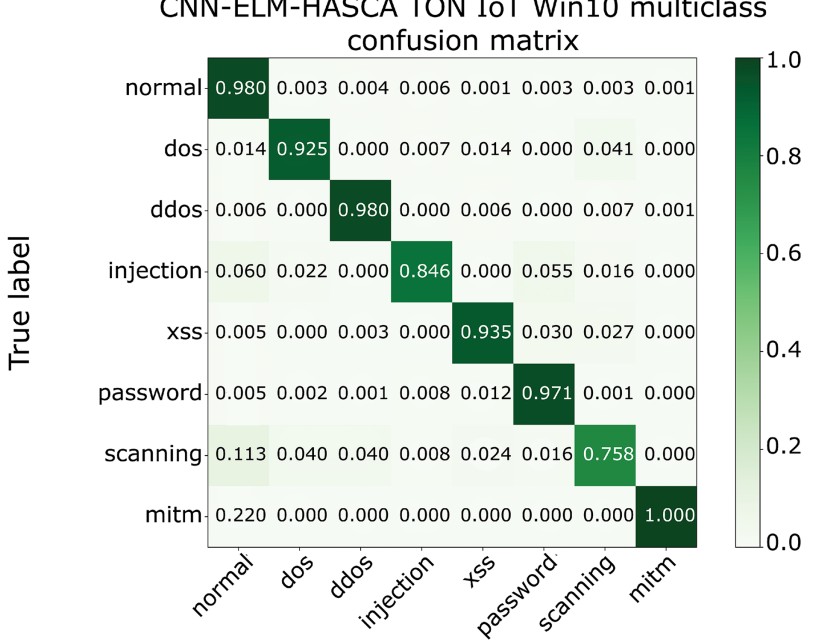

**Figure 15 The confusion matrix of the proposed CNN-ELM-HASCA method on Win 10 dataset.**

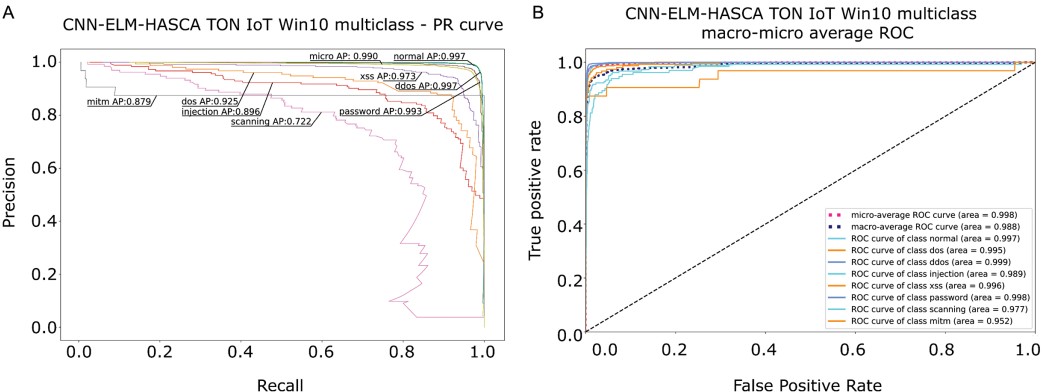

**Figure 16 Visual representations of the simulation outcomes in terms of PR AUC (A) and ROC AUC (B) curves of the proposed CNN-ELM-HASCA method on Win 10 dataset.**

values, and also standard deviation and variance values, leaving behind CNN-ELM-ChOA and CNN-ELM-SCA models.

Table 8 encapsulates the comprehensive metrics attained by the best run of every regarded model. It is worth noting that the suggested CNN-ELM-HASCA model once more attained the superior accuracy level of 96.65%, leaving behind CNN-ELM-ChOA the second best accuracy of 96.64% and CNN-ELM-SCA that acquired the third best accuracy of 96.62%. The suggested CNN-ELM-HASCA method has also displayed supremacy when

**Table 9 Shapiro–Wilk values with respect Win 7 and Win 10 problem instances for testing normality requirement.**

| Methods | HASCA | SCA | ABC | BA | WOA | EHO | ChOA | RSA |
|---|---|---|---|---|---|---|---|---|
| Win 7 | 0.371 | 0.198 | 0.205 | 0.264 | 0.337 | 0.341 | 0.406 | 0.297 |
| Win 10 | 0.391 | 0.241 | 0.193 | 0.245 | 0.305 | 0.329 | 0.287 | 0.297 |

**Table 10 Shapiro-Wilk executed over the mean differences among two samples to verify the precondition for paired-t test, followed by the paired-t test scores with respect to Win 7 and Win 10 problems**

| Methods *vs*. HASCA | SCA | ABC | BA | WOA | EHO | ChOA | RSA |
|---|---|---|---|---|---|---|---|
| **Shapiro–Wilk** | | | | | | | |
| Win 7 | 0.191 | 0.225 | 0.232 | 0.239 | 0.221 | 0.203 | 0.198 |
| Win 10 | 0.191 | 0.239 | 0.224 | 0.199 | 0.205 | 0.193 | 0.205 |
| **Paired-t test** | | | | | | | |
| Win 7 | 0.033 | 0.026 | 0.034 | 0.028 | 0.032 | 0.037 | 0.035 |
| Win 10 | 0.071 | 0.036 | 0.043 | 0.042 | 0.039 | 0.076 | 0.042 |

taking into account other statistical categories, as it attained the best results for the majority of other indicators.

Visualisations of the simulation outcomes attained on the Windows 10 dataset have been provided in Figs. 13 and 14. Figure 13 exposes the convergence graphs, box and violin plots of all noted algorithms with respect to the fitness function (Cohen's kappa in experiments) and classifying error rate. The convergence graphs again exhibit the supremacy of the HASCA method, where it is obvious that employed switching mechanism betwixt SCA and FA search processes significantly aids to the fast converging capabilities of the suggested approach. Additionally, Fig. 14 exhibits the swarm plots making possible to estimate the diversity of solutions during the last iteration of the best run for every noted method, with respect to the fitness function and classification error rate. Once more, one can take notice that each one of the solutions during the last iteration of HASCA run have been closely placed in the proximity of the best individual. Finally, Fig. 14 also provides KDE diagrams, showing that the simulation results belong to the normal distribution.

To further visualize the performance of the proposed CNN-ELM-HASCA approach on Win 10 dataset, Fig. 15 displays the confusion matrix, while Fig. 16 depicts PR AUC and ROC AUC curves achieved by the suggested method.

## Statistical tests and model validation

In order to verify the experimental outcomes and establish if they are significant from the statistical point of view, the best results from every of the fifteen executions of each considered algorithm with respect to both considered problem cases (Win 7 and Win 10 datasets) were collected and investigated as data series. Nevertheless, at the beginning, it is

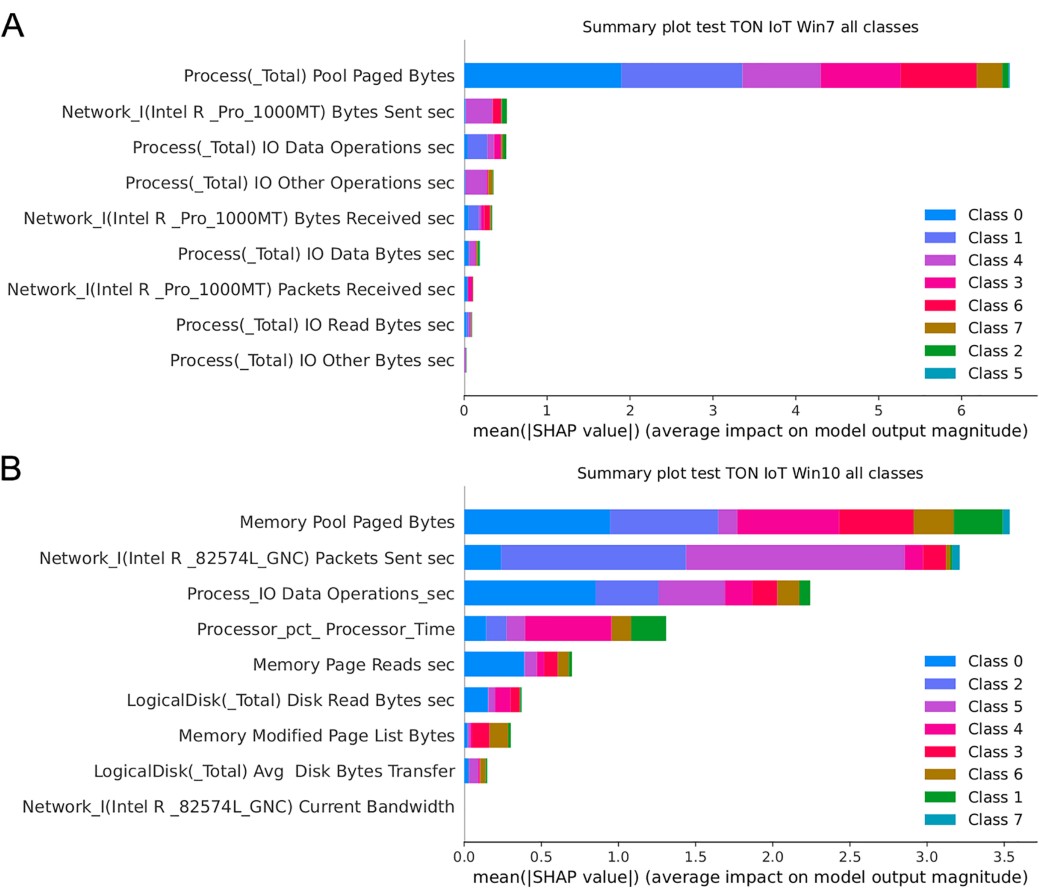

**Figure 17** SHAP summary plots for Win 7 (A) and Win 10 datasets (B).

required to establish which type of the statistical tests is appropriate—parametric or non-parametric. Prior to deciding to utilize the non-parametric tests, it is required to check if it is possible to use the parametric tests, by examining independence, normality, and homoscedasticity of the data variances (*LaTorre et al., 2021*). The first requirement, namely independence, is fulfilled as each execution of all algorithms starts by generating a set of pseudo-random variables. The second requirement, homoscedasticity, has been validated by performing Levene's test (*Glass, 1966*), and as the *p–value* of 0.67 was determined in every case, one can conclude that the homoscedasticity requirement has also been satisfied.

For verification of the last, normality requirement, Shapiro-Wilk single problem analysis has been applied (*Shapiro & Francia, 1972*). Shapiro–Wilk *p–values* have been established independently with respect to each of the regarded algorithms. The determined *p–values* for each approach were larger than 0.05, allowing the conclusion that the H0 hypothesis can not be rejected for both $alpha = 0.05$ and $alpha = 0.1$. Consequently, it means that the observed results are originating from the normal distribution. It was possible to come to the similar conclusion by simply looking at the KDE diagrams in Figs. 10 and 14. The Shapiro–Wilk testing values are given in Table 9.

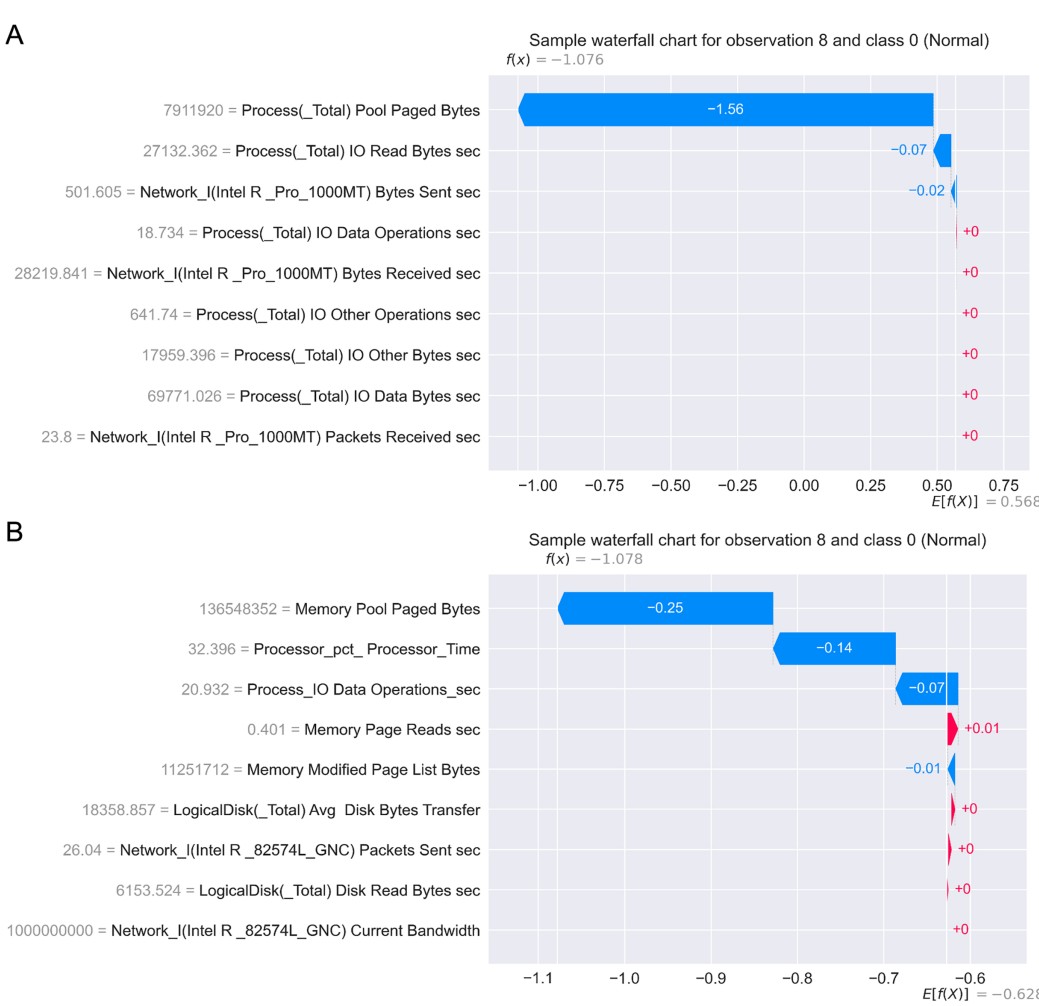

**Figure 18 SHAP waterfall plots for Win 7 (A) and Win 10 datasets (B).**

As the normality condition is fulfilled as well, it is allowed to safely utilize the parametric test. Within this manuscript, the paired-t test has been employed (*Hsu & Lachenbruch, 2014*), since it is a common choice when metaheuristics-based algorithms are evaluated (*Chen et al., 2014*). Paired-t test may be employed in case if the collection of data values can be observed as paired measurements, where the distribution of differences betwixt the pairs is required to follow the normal distribution as well. Simply put, differences among samples of each pair of metaheuristics should be normally distributed. Aiming to examine this, the absolute differences among distributions of the suggested method and other contenders were determined, followed by another application of the Shapiro–Wilk on each absolute difference. The outcomes of the Shapiro–Wilk test have shown that the *p–values* in all instances were larger than threshold value 0.05, allowing the conclusion that it is not possible to reject H0 hypothesis for $alpha = 0.05$, which means that the observed values are belonging to the normal distribution. As this is the precondition for execution of the

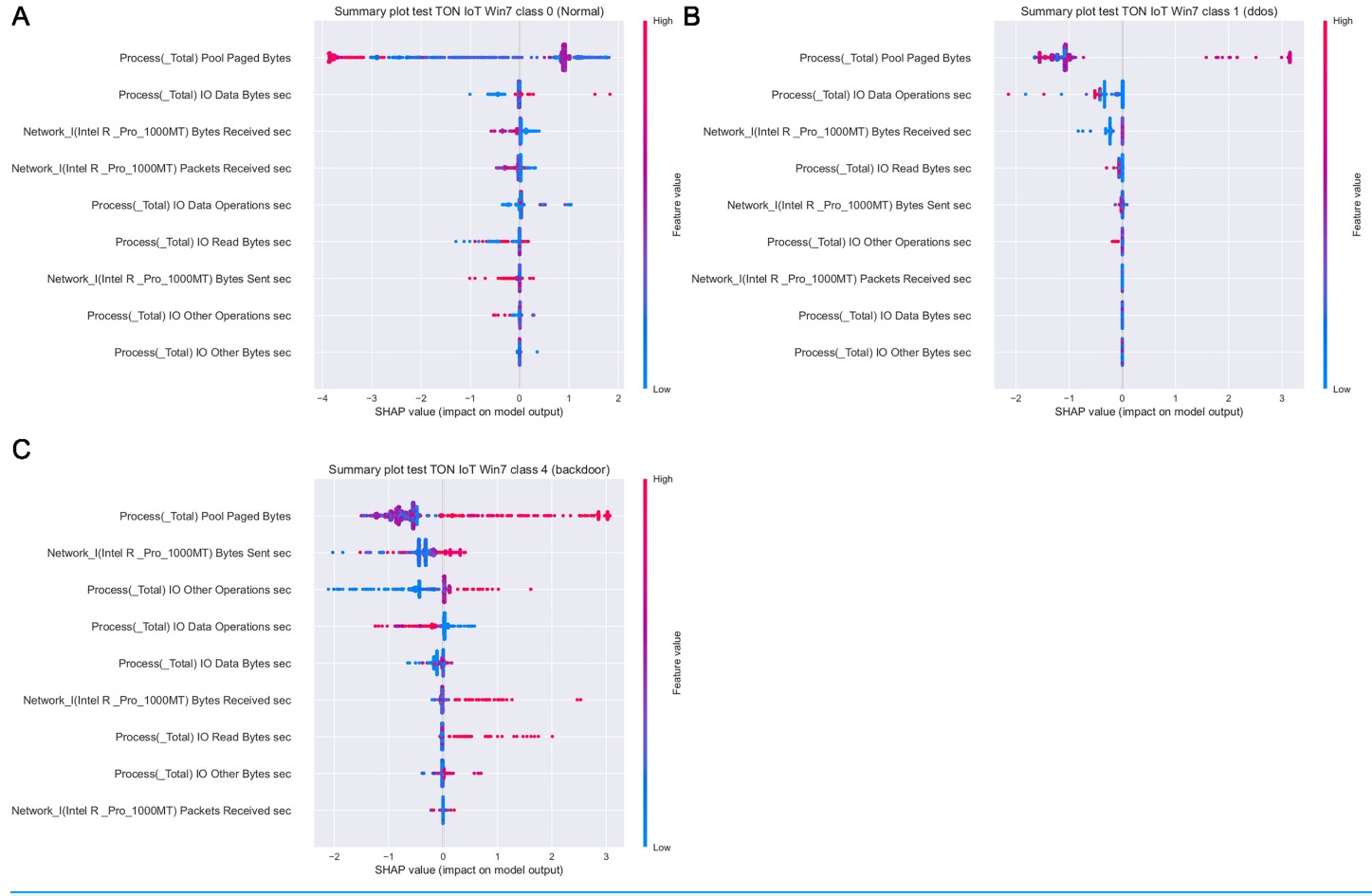

**Figure 19 SHAP summary plots for classes 0 (A), 1 (B) and 4 (C) for Win 7 experiments.**

paired-t test, it means that paired-t test can be safely used, comparing the suggested method to each and every one of contenders.

The outcomes of Shapiro–Wilk *p–values* established on the differences between the suggested approach and other contenders, followed by the paired-t test outcomes are shown in Table 10. In case of the paired-t test, the *p–values* are smaller then 0.05 for all algorithms excluding SCA and ChOA with respect to the Win 10 dataset (0.071 with SCA and 0.076 with ChOA). Accordingly, it can be established that the introduced HASCA method is significantly superior over all contenders for threshold $alpha = 0.1$, and significantly superior than all contenders excluding SCA and ChOA algorithms, when the threshold value $alpha = 0.05$ is observed.

One of the most important tasks when analysing the results of the machine learning models is to interpret them properly, aiming to discover what are the most influential features with respect to the target variable. Proper interpretation will allow the decision makers to decide more confidently, and that can be vital in the network security area. To explain the behavior of model observed in this research, the advanced explainable AI

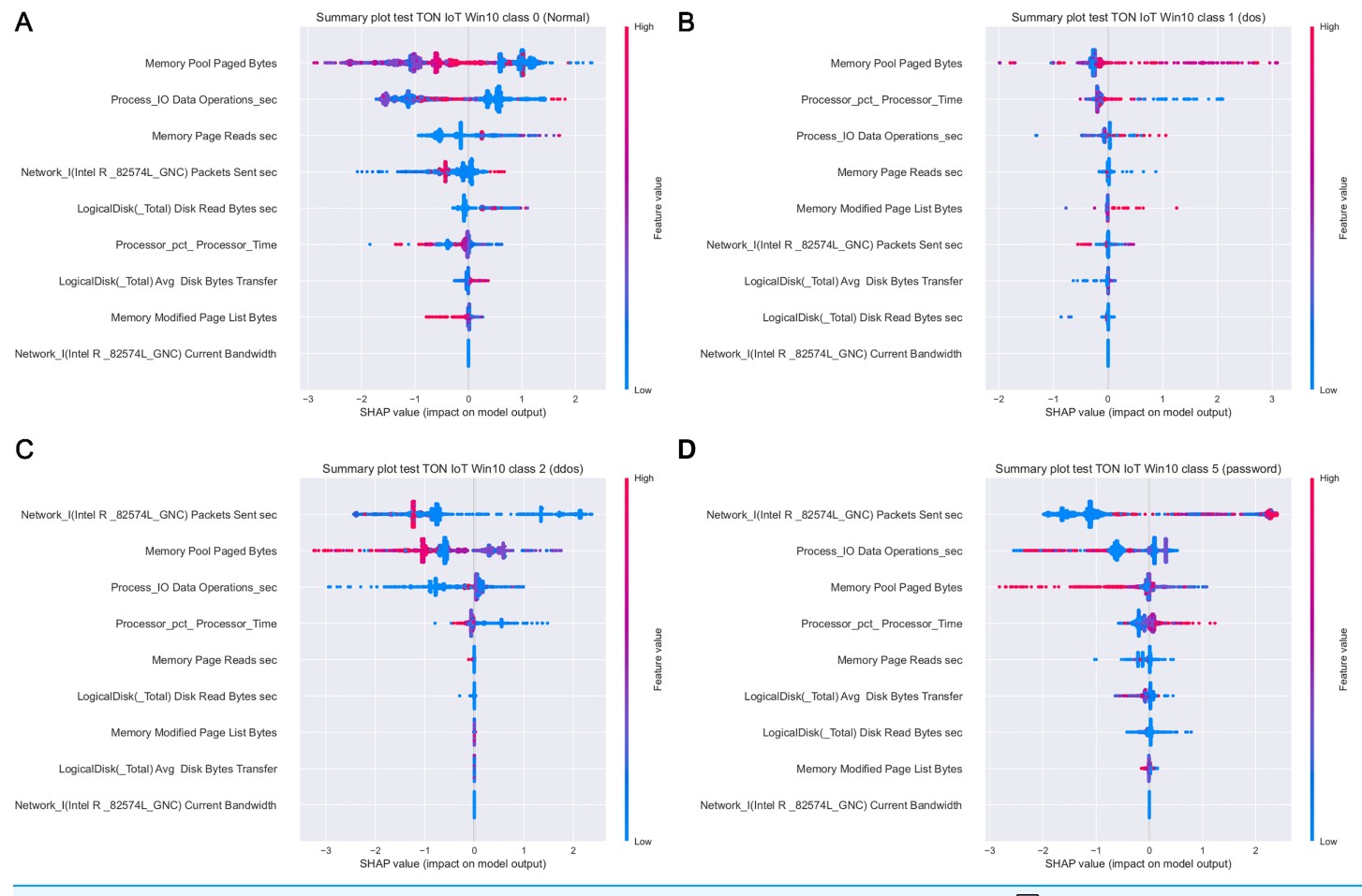

**Figure 20** SHAP summary plots for classes 0 (A), 1 (B), 2 (C) and 5 (D) for Win 10 experiments.

method SHAP was utilized, allowing the better understanding of the simulation results. SHAP procedure allows easy and fitting interpretations of the predictions made by the observed model, by measuring the importance of each feature, inspired by the game theory (*Lundberg & Lee, 2017*).

Simply said, Shapley collection of values represents the distributed payouts betwixt the features, with respect to the every feature's contibution towards the joint payout (denoting the prediction in this case). Finally, SHAP method supplements each feature with importance indicator, that measures the contribution of every one of the features on the specific forecast.

Figure 17 presents the SHAP summary plots allowing to analyse the influence of features on output classes, for both Windows 7 and Windows 10 dataset. Moreover, Fig. 18 provides simple SHAP waterfall plots, showing the extent of features affecting observation 8 with respect to class 0 (normal).

Figure 19 depicts how features influence class 0 (normal traffic), class 1 (dos) and class 4 (backdoor), with respect to the experiments with Windows 7 dataset. Similarly, 20 displays

the effect of features on class 0 (normal traffic), class 1 (dos), class 2 (ddos) and class 5 (password) for the experiments with Windows 10 dataset.

Intriguingly, given SHAP visualizations indicate that the last feature (in both Win 7 and Win 10 experiments) almost has no influence whatsoever, concluding that it may be removed based on the SHAP analysis. This feature was not removed according to *Moustafa et al. (2020)*, however, this is an important observation.

Moreover, as examples presented in Figs. 19 and 20 indicate, the analysis of the effect each feature has on a specific target can be performed. For instance, increasing the value of Memory.Pool.Paged Bytes attribute will also add to the influence that the target in that particular case will be class 1 (dos attack). Similarly, if the attribute Network_I.Intel. R82574L_GNC.Packets Sent.sec is decreased, showing the speed of sending packets through the network interface, will increase the effect on the outcome to be classified as DDOS attack.

Aiming to determine the most important attributes, it is possible to conclude that the largest influence in case of the Win 7 dataset have Process Pool Paged Bytes, Network I. Intel.R Pro 1000MT.Bytes Sent sec, and Process.Total IO Data Operations sec features. With respect to Win 10 dataset, the most important attributes are Memory.Pool.Paged Bytes, Network_I.Intel.R82574L_GNC.Packets Sent.sec, and Process.Total IO Data Operations sec. Generally speaking, the SHAP analysis clearly indicates that the possibility of the real network attack is high if the problems such as the increased virtual memory utilization and physical memory paging, or reduced speed of read and write procedures during I/O operations have been noticed. This is also in accordance to the real world experience, as it was confirmed in practice countless times.

## CONCLUSION

The intrusion detection problem in IoT networks is crucial, as unauthorized access or compromised data could lead to leaking of private information, reputation loss or even human casualties. To address this problem and keep IoT network secure, it is necessary to quickly and consistently differentiate between the malicious actions and regular activities. The research presented in this manuscript introduces a novel hybrid intrusion detection structure that can be utilized for this crucial task. The suggested approach is relying on the novel HASCA method, that was developed by modifying the elementary SCA metaheuristics and incorporating the FA search mechanism. The basic SCA algorithm has a powerful exploration, however, it does not have sufficient exploitation capabilities. Creating a low-level hybrid with FA that is known for the strong exploitation seems like a logical choice, where the advantages of both algorithms could mutually overcome their respective drawbacks. The HASCA algorithm begins execution by using the basic SCA search mechanism, however, in later stages, it is alternating betwixt SCA and FA search procedures, to enhance the exploitation.

This novel HASCA metaheuristics was used within the hybrid ML framework, that consists of the lightweight CNN and ELM model, where HASCA was used to tune the ELM's structure (number of neurons in its single hidden layer), as well as in determining weights and biases between neurons. Introduced framework was entitled CNN-ELM-

HASCA, and its performance was validated on two intrusion detection benchmark instances (Win 7 and Win 10 datasets). The attained experimental outcomes were compared to the results achieved by seven other contending metaheuristics algorithms, tested as part of the identical framework, and utilized to tune the ELM for the observed task. The proposed CNN-ELM-HASCA attained superior level of accuracy of 98.67% and 96.65% over Win 7 and Win 10 datasets, respectively.

However, as in any other research work, the proposed study outlines some limitations. First of all, the CNN structure used for feature extraction was determined manually, by 'trial and error' approach. Automatic evolving of CNN's structure (set of hyperparameters' values), *e.g.*, by utilizing metaheuristics, is very resource-intensive and it would require additional time and computing resources. However, this could be a promising topic for future research in the area. Secondly, introduced framework was evaluated on only two multi-class datasets and thus, it may require further evaluation on a wider set of benchmarking data. Finally, the SCA metaheuristics could also be further improved by investigating hybridization with other metaheuristics that show good exploitation abilities.

Regardless of above mentioned limitations, experimental outcomes presented in this study are very encouraging, and the future experiments will be focused on gaining further confidence into the suggested CNN-ELM-HASCA model. This will include validation on supplementary real-world datasets, prior to possible implementation as a part of the real IDS. Also, a further research may turn towards tuning of the CNN structure along with the ELM for this important challenge as a part of the two-level framework—CNN tuning in the first layer and the ELM optimization in the second one. Additionally, due to the fact that there is still a significant research gap in this domain, with numerous ML/DL models, and with metaheuristics available in the modern literature, future research may also be focused on experimentation with various ML/DL models and metaheurisitcs combinations for significant IoT security challenge.

### Funding
This research is funded by the Universiti Kebangsaan Malaysia (Grant code: GUP-2022-060). Nor Samsiah Sani and Maifuza Mohd Amin provided funding for the proposed research. The funders had no role in study design, data collection and analysis, decision to publish, or preparation of the manuscript.

### Grant Disclosures
The following grant information was disclosed by the authors:
Universiti Kebangsaan Malaysia: GUP-2022-060.

### Competing Interests
The authors declare that they have no competing interests.

## Author Contributions

- Milos Dobrojevic performed the experiments, analyzed the data, performed the computation work, authored or reviewed drafts of the article, and approved the final draft.
- Miodrag Zivkovic conceived and designed the experiments, prepared figures and/or tables, authored or reviewed drafts of the article, and approved the final draft.
- Amit Chhabra conceived and designed the experiments, analyzed the data, performed the computation work, prepared figures and/or tables, and approved the final draft.
- Nor Samsiah Sani performed the experiments, analyzed the data, authored or reviewed drafts of the article, and approved the final draft.
- Nebojsa Bacanin conceived and designed the experiments, analyzed the data, performed the computation work, prepared figures and/or tables, authored or reviewed drafts of the article, and approved the final draft.
- Maifuza Mohd Amin performed the experiments, performed the computation work, authored or reviewed drafts of the article, and approved the final draft.

## Data Availability

The dataset is available in the Supplemental Files and at GitHub and Zenodo: https://github.com/nbacanin/Peerj2023.

Milos Dobrojevic, Miodrag Zivkovic, Amit Chhabra, Nor Samsiah Sani, Nebojsa Bacanin, & Maifuza Mohd Amin. (2023). Hybrid Machine Learning Model Tuned by Enhanced Sine Cosine Metaheuristics and Model Interpretation based on SHAP Approach. https://doi.org/10.5281/zenodo.7725513.

## Supplemental Information

Supplemental information for this article can be found online at http://dx.doi.org/10.7717/peerj-cs.1405#supplemental-information.

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
