# Peer review of "Addressing Internet of Things security by enhanced sine cosine metaheuristics tuned hybrid machine learning model and results interpretation based on SHAP approach"

_PeerJ Computer Science, doi:10.7717/peerj-cs.1405_

## Round 0.1 · original submission · Major Revisions

I have completed my evaluation of your manuscript. The reviewers recommend reconsideration of your manuscript following major revision. I invite you to resubmit your manuscript after addressing the comments below.

Reviewer 1 ·

Basic reporting

The paper falls within the scope of the journal. The authors have performed useful research and have tried to show the benefits of the proposed approach. The paper has potential, but must be improved. My comments are as follows:
- The abstract is too extensive, Should be more concise with only core and standard elements of writing the abstract. Unnecessary descriptions should be removed.
- Figure shouldn't be part of the introduction. Please move Figure 1 to another section.
- The following papers can be cited: Biliavska, V, Castanho, R. A., & Vulevic, A. (2022). Analysis of the Impact of Artificial Intelligence in Enhancing the Human Resource Practices, J. Intell. Manag. Decis., 1(2), 128-136. https://doi.org/10.56578/jimd010206
Adedotun, A. F. (2022). Hybrid Neural Network Prediction for Time Series Analysis of COVID-19 Cases in Nigeria, J. Intell. Manag. Decis., 1(1), 46-55. https://doi.org/10.56578/jimd010106
- Figure 2 is well known and I think that is redundant in this paper.

Experimental design

- Equations are well explained and represented.
- All necessary information has been provided.
- Figure 3 isn't readable. Please improve it. Also, Figure 4 should be more quality.

Validity of the findings

Generally, this part of the paper is well elaborated with enough explanation. Besides some minor corrections should be done.
- Figure 6 isn't readable.
- Figures 9 and 10 can be converted in Table. Now isn't readable.

Additional comments

- Please ensure discussion with implications, limitations etc.
- Future research, in conclusion, is poor, you should add more guidelines.
- It is mandatory to reduce self-citations. Now, this number is about 20. Should be reduced to a number less than 10.

Reviewer 2 ·

Basic reporting

The title should clearly indicate the application of the model – area IoT security.
The summary and conclusion correspond to the essence of the work.
The results are presented both tabularly and graphically in an adequate manner.
The acronyms are consistently explained when first used in the text, except in line 145 - FS.
The use of terminology is correct and it complies with applicable standards.

Experimental design

In lines 183-184, the authors said - The extensive literature survey has also shown that this particular CNN and ELM hybrid combination has never been utilized to address the intrusion detection problem.
Please briefly explain the reasons (motive) that directed you towards the joint use of these two methods (good and bad sides of the methods and what was the assumption that you would achieve the combination), and whether there were considered other possibilities.
I suggest the authors provide a brief comment on the correlation matrices (Figure 9 and Figure 10).
The authors clearly explained the methodology and method of literature selection and presented it clearly in the text, graphically, and tabularly. The way of presenting the results graphically and the choice of graphic types for a clearer visualization of the results are excellent, except that the marks within the graphics are barely visible

Validity of the findings

In the introductory part, the authors indicated the problem and clearly explained the existing ways of solving it. The proposed model is explained in detail, its relevance is compared with other models, and results are presented that unequivocally indicate the applicability of the model.
I request the authors to point out the limitations of the model's application in the concluding part.

Additional comments

no comment

Reviewer 3 ·

Basic reporting

The authors carried out a valuable research in the field of cyber security and optimization algorithms. The developed HASCA algorithm was compared to the scores acquired by 7 additional metaheuristics, separately implemented for the purpose of this research. The chosen algorithms were the basic SCA, ABC, bat algorithm (BA), whale optimization algorithm (WOA), elephant herding optimization (EHO), chimp optimization algorithm (ChOA) and reptile search algorithm (RSA).
The article includes an appropriate introduction and background to demonstrate how the work fits into the broader field of knowledge.
I suggest the following improvements:
There are no explanation regarding the abbreviations in the text, for example CNN, ELM, etc. are introduced without explaining what it stands for.
There is excessive self-citation of the authors’ papers. If possible, it should be reduced.

Experimental design

The research question is well-defined, relevant and meaningful. It is explained how the research fills an identified knowledge gap.
The research is conducted in conformity with the prevailing ethical standards in the field. Methods are described with sufficient detail.

Validity of the findings

The obtained results are of interest to the general and academic audience. The conclusions are appropriately stated and connected to the original investigated question.

---

## Round 0.2 · Minor Revisions

Please address the remaining comments. Please only add references if you feel they are relevant to your work.

Reviewer 1 ·

Basic reporting

- The abstract is too extensive, Should be more concise with only core and standard elements of writing the abstract. Unnecessary descriptions should be removed. The authors made appropriate changes.
- Figure shouldn't be part of the introduction. Please move Figure 1 to another section. The authors made appropriate changes.
- The following papers can be cited: Biliavska, V, Castanho, R. A., & Vulevic, A. (2022). Analysis of the Impact of Artificial Intelligence in Enhancing the Human Resource Practices, J. Intell. Manag. Decis., 1(2), 128-136. https://doi.org/10.56578/jimd010206
Adedotun, A. F. (2022). Hybrid Neural Network Prediction for Time Series Analysis of COVID-19 Cases in Nigeria, J. Intell. Manag. Decis., 1(1), 46-55. https://doi.org/10.56578/jimd010106. The authors cited the suggested references.
- Figure 2 is well known and I think that is redundant in this paper. Figure has been removed.

Experimental design

- Equations are well explained and represented.
- All necessary information has been provided.
- Figure 3 isn't readable. Please improve it. Also, Figure 4 should be more quality. Solved.

Validity of the findings

Generally, this part of the paper is well elaborated with enough explanation. Besides some minor corrections should be done.
- Figure 6 isn't readable.
- Figures 9 and 10 can be converted in Table. Now isn't readable

The authors made appropriate changes.

Additional comments

- Please ensure discussion with implications, limitations etc.
- Future research, in conclusion, is poor, you should add more guidelines.
- It is mandatory to reduce self-citations. Now, this number is about 20. Should be reduced to a number less than 10.

The authors made appropriate changes.

Reviewer 2 ·

Basic reporting

no comment

Experimental design

no comment

Validity of the findings

no comment

Additional comments

I think that the authors have very conscientiously and precisely responded to the comments and suggestions of the reviewers and thereby improved the quality of their paper.

Reviewer 3 ·

Basic reporting

The revised manuscript has been improved accordingly.

Experimental design

The revised manuscript has been improved accordingly.

Validity of the findings

The revised manuscript has been improved accordingly.

---

## Round 0.3 · accepted · Accept

I am accepting the manuscript for publication.